# The Impact of Transport Model Differences on $CO_2$ Surface Flux Estimates from OCO-2 Retrievals of Column Average $CO_2$

Sourish Basu[1,2], David F. Baker[1,3], Frédéric Chevallier[4], Prabir K. Patra[5], Junjie Liu[6], and John B. Miller[1]

[1]NOAA Earth System Research Laboratory, Global Monitoring Division, Boulder CO, USA
[2]Cooperative Institute for Research in Environmental Sciences, University of Colorado, Boulder CO, USA
[3]Cooperative Institute for Research in the Atmosphere, Colorado State University, Ft. Collins CO, USA
[4]LSCE-CEA-UVSQ-CNRS, Orme des Merisiers, Gif-sur-Yvette, France
[5]RCGC/IACE/ACMPT, Japan Agency for Marine-Earth Science and Technology (JAMSTEC), Yokohama, Japan
[6]Jet Propulsion Laboratory, California Institute of Technology, Pasadena CA, USA

*Correspondence to:* Sourish Basu (sourish.basu@colorado.edu)

**Abstract.** We estimate the uncertainty of $CO_2$ flux estimates in atmospheric inversions stemming from differences between different global transport models. Using a set of Observing System Simulation Experiments (OSSEs), we estimate this uncertainty as represented by the spread between five different state-of-the-art global transport models (ACTM, LMDZ, GEOS-Chem, PCTM and TM5), for both traditional in situ $CO_2$ inversions as well as inversions of $XCO_2$ estimates from the Orbiting Carbon Observatory 2 (OCO-2). We find that in the absence of relative biases between in situ $CO_2$ and OCO-2 $XCO_2$, OCO-2 estimates of terrestrial flux for TRANSCOM-scale land regions can be more robust to transport model differences compared to corresponding in situ $CO_2$ inversions. This is due to a combination of the increased spatial coverage of OCO-2 samples and the total column nature of OCO-2 estimates. We separate the two effects by constructing hypothetical in situ networks with the coverage of OCO-2 but with only near-surface samples. We also find that the transport-driven uncertainty in fluxes is comparable between well-sampled northern temperate regions and poorly sampled tropical regions. Furthermore, we find that spatiotemporal differences in sampling, such as between OCO-2 land and ocean soundings, coupled with imperfect transport, can produce differences in flux estimates that are larger than flux uncertainties due to transport model differences. This highlights the need for sampling with as complete a spatial and temporal coverage as possible (e.g., using both land and ocean retrievals together for OCO-2) to minimize the impact of selective sampling. Finally, our annual and monthly estimates of transport-driven uncertainties can be used to evaluate the robustness of conclusions drawn from real OCO-2 and in situ $CO_2$ inversions.

## 1 Introduction

Atmospheric measurements of $CO_2$ show that on average, half of the anthropogenic emissions of $CO_2$ are taken up each year by the land and oceans (Ballantyne et al., 2012). Allocating this global sink to specific regions, or even partitioning it between land and oceans, has proved challenging (Schimel et al., 2014). Understanding the mechanisms behind this allocation, and their response to climate variability, is crucial for accurately estimating the carbon cycle impact on future climate scenarios

(Friedlingstein et al., 2014). Current approaches to quantify the spatial distribution and temporal variation of carbon sources and sinks can be broadly classified into two categories, "top down" and "bottom up". Bottom up methods, such as biosphere models and ocean biogeochemistry models, calculate the surface exchange of $CO_2$ between two reservoirs by modelling the physical processes in the reservoirs that lead to such exchanges. Top down methods, generally speaking, infer surface fluxes of
$CO_2$ from measured spatiotemporal gradients in tracer concentrations in either reservoir.

The most common top down method for estimating surface fluxes of $CO_2$ from atmospheric measurements is an atmospheric inversion. An inversion infers surface fluxes from observed spatiotemporal gradients of $CO_2$ in the atmosphere by simulating atmospheric transport to connect the two. Most inversions are Bayesian in nature, in that they calculate corrections from a prior flux scenario (typically from bottom up models) under constraints of assumed errors in the prior fluxes and atmospheric
measurements. The flux estimates from an inversion, therefore, are subject to the assumed prior flux map and its error structure, the atmospheric transport model, the set of atmospheric observations assimilated, and the assimilation technique. Due to the diversity of each of these elements in the current suite of atmospheric inversions, estimates of $CO_2$ fluxes from biomes and ocean basins vary widely across inversions, even though they agree on the global $CO_2$ budget (Peylin et al., 2013), as would be expected from mass balance considerations.

Peylin et al. (2013) showed that the northern extra-tropical sink was fairly consistent across inversions of in situ $CO_2$ data, but the partitioning between the tropics and the southern extra-tropics was more variable. The tropics were found to be responsible for most of the interannual variability of the global $CO_2$ growth rate, and northern Asia was found to be responsible for an increasing northern land carbon uptake between 1990 and 2008. However, the tropics and northern Asia were also the regions most severely under-sampled by the surface $CO_2$ observation network used by the inversions in Peylin et al. (2013). Therefore,
it remained an open question whether their conclusions were real or artifacts of insufficient observational constraints.

Satellite estimates of atmospheric $CO_2$ mole fraction, in principle, can add observational constraints over remote areas that are difficult to sample with surface sampling sites, such as the tropics, Boreal Eurasia, and much of the oceans. This was the chief motivation behind the Greenhouse gases Observing SATellite (GOSAT), launched in 2009 (Kuze et al., 2009). GOSAT near infrared (NIR) spectra of reflected sunlight have been analyzed to estimate the column average $CO_2$ mole fraction under
its orbit. It was hoped that these column averages – hereafter called $XCO_2$ – assimilated by atmospheric inversions, would help constrain the $CO_2$ flux over regions such as the tropics and northern Asia. Houweling et al. (2015) showed that assimilating GOSAT $XCO_2$ indeed reduced the spread in tropical land flux estimates across a suite of atmospheric inversions. However, the year-round coverage of GOSAT did not extend beyond $\pm 36°$ latitude, limiting its ability to draw conclusions about high latitude fluxes. Over the tropics, despite the year-round coverage, GOSAT retrievals were sparse due to cloud cover and high
aerosol loading from biomass burning, also limiting its ability to constrain tropical fluxes. The balance between tropical and temperate fluxes estimated from GOSAT soundings was also inconsistent with information from independent aircraft profiles, raising questions about its validity (Houweling et al., 2015).

In 2014, the next $CO_2$ observing satellite, Orbiting Carbon Observatory 2 (OCO-2), was launched (Crisp et al., 2017; Eldering et al., 2017). Compared to GOSAT, OCO-2 has more extensive spatial coverage, both in the density of soundings
as well as their latitudinal extent. Its higher measurement signal to noise allows for higher precision retrievals of $XCO_2$, and

higher spatial sampling density enables easier validation with the ground-based Total Carbon Column Observing Network or TCCON (Wunch et al., 2017). OCO-2 also has a smaller footprint compared to GOSAT, potentially enabling more retrievals over the tropics by looking through gaps in clouds, over scenes that GOSAT might have treated as cloud-contaminated. Due to the more extended spatial coverage, higher sampling density, higher precision and better validation opportunity, OCO-2 can potentially provide better constraints on surface $CO_2$ fluxes than what has hitherto been possible from the surface network and GOSAT. Several inverse modelling groups are currently engaged in investigating this potential.

One of the key problems in estimating $CO_2$ fluxes from GOSAT retrievals is the presence of small but spatially coherent biases in the retrievals arising from, e.g., a dependence of the retrieved $XCO_2$ on aerosols or surface albedo (Cogan et al., 2012; Guerlet et al., 2013; Wunch et al., 2011). Some synthetic data studies such as Chevallier et al. (2007) had warned that such sub-ppm biases might significantly reduce the utility of satellite $XCO_2$ retrievals, but most earlier studies either did not consider this complication (Rayner and O'Brien, 2001; Hungershoefer et al., 2010) or claimed that it was easily fixable (Miller et al., 2007). In practice, these biases were found to strongly affect estimated fluxes in atmospheric inversions of GOSAT data (e.g., Basu et al., 2013; Feng et al., 2016). Initial analyses suggest that OCO-2 estimates of $XCO_2$ likely suffer from similar biases (Wunch et al., 2017), although they can be better characterised due to the increased density of soundings. Efforts are underway to characterize and remove such biases through improvements in the radiative transfer and surface reflectance models. Current validation strategies for satellite $XCO_2$ have their own limits, since their truth metrics (e.g., TCCON $XCO_2$) may not be sufficiently accurate (Basu et al., 2011). Therefore, as satellite retrieval algorithms achieve higher accuracy, they will need better validation strategies in the future. It is likely that with further progress in those directions, $XCO_2$ biases will go down to the point where they no longer limit our ability to infer regional $CO_2$ fluxes.

Even with completely unbiased $XCO_2$ retrievals, surface flux estimates would still be subject to uncertainties related to the atmospheric transport model, the optimization technique employed, and the balance between data and prior flux errors. At present, it is not clear whether the divergence in flux estimates seen in intercomparisons such as Houweling et al. (2015) is driven primarily by the variety of $XCO_2$ retrievals assimilated or the other factors mentioned above, although more limited intercomparisons suggest that those other factors may be at least as important as the differences in $XCO_2$ assimilated (Chevallier et al., 2014). It is possible that the uncertainty in a regional flux estimate stemming from factors specific to the inverse modelling setup is larger than what we can tolerate for detecting, say, the climate impact on those fluxes. In that case, even perfectly accurate estimates of satellite-based $XCO_2$ will not enable us to answer the carbon cycle questions we hope to answer with current and future $CO_2$ sensing satellite missions. It is therefore crucial that we quantify the impact of factors specific to an inverse modeling setup on the uncertainty of inferred surface fluxes.

In this study, we consider one of those factors, namely the atmospheric transport model. Using a series of Observing System Simulation Experiments (OSSEs), we quantify the uncertainty in flux estimates due to differences between present day state-of-the-art atmospheric transport models. The approach is similar to that used by earlier work (Chevallier et al., 2010; Houweling et al., 2010; Locatelli et al., 2013). To wit:

1. From a common set of surface fluxes (henceforth called "true" fluxes), we use a suite of different atmospheric transport models to produce a suite of time-varying three-dimensional atmospheric $CO_2$ fields.

2. We sample these fields to produce synthetic observations of $CO_2$ at in situ and OCO-2 sampling locations.

3. We assimilate these synthetic observations in a single data assimilation system with a single transport model.

4. For a given data stream (e.g., in situ observations, or OCO-2 land nadir), the spread in the posterior fluxes is an estimate of the uncertainty driven by transport model differences.

In earlier work, Chevallier et al. (2010) performed their analysis for the GOSAT instrument, while Houweling et al. (2010) focussed on the (planned) A-SCOPE active sensor. Our methodology is closest to that of Locatelli et al. (2013), who estimated the transport model driven uncertainty of $CH_4$ fluxes assimilating only surface layer data. In our analysis, we try to answer two specific questions:

1. For atmospheric inversions assimilating OCO-2 $XCO_2$ retrievals, what are the uncertainties on posterior flux estimates – at different spatiotemporal scales – that arise due to the divergence of present day state of the art atmospheric tracer transport models?

2. Are the uncertainties larger or smaller if we assimilate only in situ measurements of $CO_2$? In other words, does assimilating space-based total column $XCO_2$ such as OCO-2 $XCO_2$ magnify or diminish transport model related uncertainties in the flux estimates?

The second question stems from a long-standing hypothesis that simulating $XCO_2$ in a model is less sensitive to transport errors such as errors in the modeled planetary boundary layer (PBL), making $XCO_2$ assimilations less sensitive to transport errors than PBL $CO_2$ assimilations (Rayner and O'Brien, 2001). This is plausible, since modeling convection and the formation of the PBL are leading order uncertainties in present day transport models (Parazoo et al., 2012). Any error in modeling the exact PBL height and vertical mass flow translates into an error in estimated fluxes, if the primary assimilated data for an inversion are PBL $CO_2$ mole fractions. On the other hand, the column average $XCO_2$ is relatively insensitive to convective transport errors and the exact PBL height, so those types of transport errors may have less influence on estimated fluxes if the primary data are $XCO_2$. However, the spatiotemporal variations in $XCO_2$ due to surface fluxes are smaller than corresponding variations in PBL $CO_2$. Therefore, $XCO_2$ inversions starting from biased priors (true for most if not all current inversions) may be less accurate than PBL $CO_2$ inversions. In the net, it is not clear whether lower transport errors in modeled $XCO_2$ can compensate for lower flux signals to give us more accurate fluxes (Houweling et al., 2010; Chevallier et al., 2010).

## 2   Data and methodology

As described earlier, we ran a suite of transport models with the same boundary conditions (initial mole fraction field and surface fluxes), sampled them to produce a suite of synthetic observations, and then assimilated those observations in the same inversion framework to come up with an estimate of flux uncertainty due to transport model differences. We describe the individual elements of this process below.

## 2.1 "True" fluxes

Synoptic differences between transport models are likely correlated to surface fluxes, since they are influenced by common drivers such as temperature, precipitation and insolation. Therefore, it is important to use realistic fluxes to generate the true scenario. We produce the true surface fluxes by assimilating $CO_2$ data from the National Oceanic and Atmospheric Admin-
istration's (NOAA) Global Greenhouse Gas Reference Network (GGGRN) and the TCCON in a TM5 4DVAR atmospheric inversion (described later in § 2.4). The inversion spanned June 1, 2014 to April 1, 2016. This ensured that the true fluxes had realistic land and ocean sinks consistent with the observed global $CO_2$ growth rate. At the end of the optimization, TM5 4DVAR wrote out global $1° \times 1°$ 3-hourly total $CO_2$ fluxes for transport models to ingest in the next step.

## 2.2 Generation of $CO_2$ fields

We ran a suite of transport models between June 1, 2014 and April 1, 2016 with the true fluxes produced earlier, starting from the same initial $CO_2$ mole fraction field as the inversion used to produce the true fluxes. The suite consisted of TM5, LMDZ, ACTM, PCTM and GEOS-Chem. Details of the individual models can be found in the respective references in Table 1. It is important to note here that this suite of models spans the range of transport models currently being used by various members of the OCO-2 Science Team to assimilate OCO-2 $XCO_2$ retrievals. Moreover, these models are driven by four
different meteorological reanalysis products, ECMWF ERA Interim (TM5, LMDZ), MERRA (PCTM), MERRA2 (GEOS-Chem) and JMA-55 (ACTM). These four products span the gamut of meteorological fields used by most atmospheric inversions today. Therefore, the divergence of flux estimates seen in this study can be taken to be a reasonable measure of the divergence expected in real data inversions with these transport models.

   The transport models produced hourly (PCTM) or 3-hourly (TM5, LMDZ, ACTM, GEOS-Chem) $CO_2$ fields at their in-
dividual lateral and vertical resolutions, which are listed in Table 1. Note that the temporal granularity listed is the time step at which the $CO_2$ mole fraction field was written out; the time step of the models for calculating transport is usually smaller. The models also wrote out the geopotential heights and atmospheric pressures at the vertical layer edges. As a first check, we verified that global average $CO_2$ mole fractions from the different models, calculated from their own pressure and $CO_2$ fields, closely matched the expected time series from the true fluxes diluting into an atmosphere of $5.123 \times 10^{18}$ Kg, the total
dry air mass of TM5. Figure 1 shows these time series. It is evident that while all the colored lines are very close to the dashed black line, there are small differences that are seasonally coherent. These differences arise from differences in the molar mass of carbon assumed by the models (e.g., 12 grams/mole vs 12.01115 grams/mole), small differences in the air mass between different models and the handling of water vapor in the model atmosphere. Rather than standardize the models to remove these small differences, we decided to keep them since they reflect legitimate differences between the models that would express
themselves in real data inversions.

**Table 1.** The different atmospheric transport models run in this study to produce $CO_2$ fields.

| Model | Resolution (lon × lat) | Vertical layers | Temporal granularity | Meteorology | Reference |
|---|---|---|---|---|---|
| TM5 | $3° \times 2°$ | 25 | 3 hours | ERA Interim | Krol et al. (2005) |
| LMDZ | $3.75° \times 1.875°$ | 39 | 3 hours | ERA Interim | Hourdin et al. (2006) |
| ACTM | $1.125° \times 1.125°$ | 32 | 3 hours | JRA-55 | Patra et al. (2009) |
| PCTM | $1.25° \times 1°$ | 40 | 1 hour | MERRA | Kawa et al. (2004) |
| GEOS-Chem | $5° \times 4°$ | 47 | 3 hours | GEOS FP | Nassar et al. (2010) |

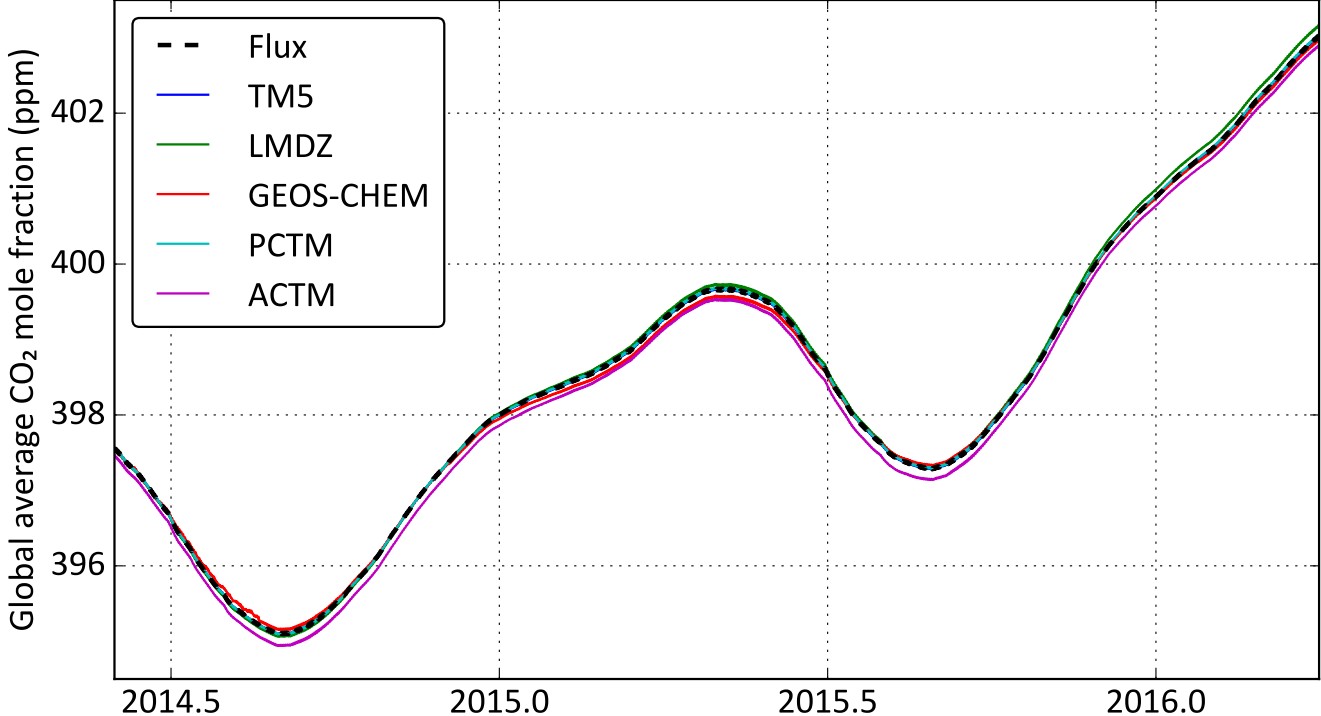

**Figure 1.** Time series of the global average $CO_2$ mole fraction expected from the true flux scenario (bold black dashed line) and calculated from the individual model outputs (colored lines). The flux scenario only provides increments of the mole fraction, so these increments were added to the initial mole fraction of TM5 to calculate the black line.

## 2.3 Generation of synthetic data

The five different modelled dry air mole fraction $CO_2$ fields were sampled with the same code to produce synthetic observations of $CO_2$ from in situ and satellite platforms. Table 2 gives the number of samples per year from each data stream, and the generation of pseudo-data are described below.

**Table 2.** Number of pseudo-observations per year from the different observing systems and sampling strategies.

| Data stream | Observations/year |
|---|---|
| MBL | 37558 |
| IS | 107963 |
| LN | 49311 |
| LG | 46103 |
| OG | 163452 |

### 2.3.1 In situ sampling

Synthetic in situ samples corresponded to the times and locations of $CO_2$ measurements at network sites maintained by NOAA and partner agencies, as contained in ObsPack versions GV 2.1 and NRT 3.2.2 (https://www.esrl.noaa.gov/gmd/ccgg/obspack/). The following data filtering was applied:

1. Campaign data from aircrafts, such as CALNEX, SONGNEX and ORCAS were excluded. In situ $CO_2$ data from the CONTRAIL program were also excluded.

2. At low altitude sites, only mid-afternoon hourly averages were used.

3. At mountain-top sites, only late night hourly averages were used.

4. For coastal sites, where the sampling protocol differentiated between background and non-background air, only background samples were used.

5. Bi-weekly to monthly NOAA aircraft profiles, mostly over North America, were included. Flask $CO_2$ data from the CONTRAIL program were also included.

Note that these filters were applied to come up with a set of sampling coordinates (locations and times) to represent realistic sampling frequency and density for real data inversions. No actual $CO_2$ measurements were used from either ObsPack version. In addition, the sampling times and locations corresponding to mid-afternoon $CO_2$ samples from six towers belonging to the Japan-Russia Siberian Tall Tower Inland Observation Network (JR-STATION) were also included in our IS network (Sasakawa et al., 2013).

Each model $CO_2$ field was sampled at these sampling coordinates, adhering as closely as possible to the sampling protocol that model would use in a real data inversion. For example, if a site's elevation places it in the lowermost model layer, TM5 samples it one layer above to avoid surface effects, while the other four models sample it in the surface layer. This distinction was kept while sampling the five models. The set of synthetic observations generated with this sampling, and corresponding flux estimates, will be referred to as "IS" in the rest of this manuscript. During this work, we discovered an artifact in our

version of PCTM at the South Pole, which was fixed by moving the South Pole site 2° north along 0° longitude (details in Appendix A).

In addition, we also considered a subset of the IS samples that corresponded closely to the network used by Baker et al. (2006). The network used in that TRANSCOM 3 model intercomparison experiment chiefly consisted of marine boundary layer and background sites, suitable for assimilation in coarse resolution flux estimation systems of the time. Since then, many continental sites have come online. These sites are located closer to terrestrial fluxes and therefore have larger flux-induced variations in the $CO_2$ mole fraction. However, modeling these variations accurately depends on modeling the continental boundary layer accurately, which is one of the most uncertain aspects of atmospheric transport modeling. By comparing the spread in our IS flux estimates to that from assimilating a more limited set of mostly background sites comparable to Baker et al. (2006), we sought to answer the question of whether the cost of increased model uncertainty in the continental PBL outweighed the benefit of more measurements from the non-background sites.

We constructed this limited subset of IS, henceforth referred to as "MBL", as follows. We subselected our IS dataset for sites that were used by Baker et al. (2006). Three sites used by Baker et al. (2006), namely CMN, GSN and HAT, did not exist in our IS dataset and therefore were not used. ITN and JBN in Baker et al. (2006) were replaced by SCT (Beech Island, South Carolina) and DRP (Drake Passage) respectively, two currently operational sites (cruises in the case of DRP) geographically nearest to the discontinued ITN and JBN. The resulting MBL network corresponded as closely as possible to the mostly background network used by Baker et al. (2006), while also reflecting changes in the $CO_2$ sampling network since then.

### 2.3.2 OCO-2 sampling

The five different model $CO_2$ fields were sampled at the locations and times of OCO-2 retrievals from the ACOS version 7r algorithm (O'Dell et al., 2012), as archived at https://disc.gsfc.nasa.gov/uui/datasets/OCO2_L2_Standard_V7r/summary. Real data inversions of OCO-2 typically only use retrievals of "good" quality, selected by `xco2_quality_flag=0`. We performed the same selection of the sounding locations to mimic realistic spatiotemporal coverage. The vertical profiles of $CO_2$ from all the models were convolved with the OCO-2 column averaging kernels and prior profiles of the corresponding real retrievals to produce sets of synthetic OCO-2 $XCO_2$. These synthetic $XCO_2$ were classified according to sounding mode and surface type of the original soundings, to come up with land nadir (LN), land glint (LG) and ocean glint (OG) synthetic OCO-2 $XCO_2$ for each transport model.

OCO-2 takes 24 samples every second, which span ∼7 km along track. Column average $CO_2$ is expected to be highly correlated over these short length scales (Worden et al., 2016), and therefore these 24 retrievals do not provide independent information about $XCO_2$. However, most trace gas inversions – including TM5 4DVAR – treat all measurements as independent. Moreover, most global transport models have grid cells hundreds of km in size, and therefore cannot model or interpret the small spatial scale $XCO_2$ variations seen by OCO-2. To avoid highly correlated measurements being treated as independent measurements in our assimilation, and to bring the spatial resolution of the retrievals more in line with the resolution of transport models used in most global inversions, we average the synthetic $XCO_2$ in 10 s bins along orbit, which results in one value per orbit per ∼70 km bin along track. The averaging is done in two steps. First, retrievals are averaged over 1 s

bins, with weights inversely proportional to the square of the posterior retrieval uncertainty for each retrieval. Next, over a 10 s interval, all 1 s bins with at least one valid retrieval are averaged to create a 10 s average. This two-step averaging is done to avoid weighting the 10 s average disproportionately towards one part of the ~70 km track which might have a lot of retrievals. Soundings of different modes (LN, LG or OG) are averaged separately to create different 10 s averages for each mode. OCO-2

averaging kernels and prior profiles are similarly averaged to create 10 s mean averaging kernels and prior profiles.

### 2.3.3 In situ sampling at OCO-2 sounding locations

The difference between OCO-2 and in situ samples are two fold, (i) the first is a column measurement while the second is a point measurement, and (ii) the spatiotemporal coverages of the two systems are vastly different. Differences between OCO-2 and in situ inversions convolve the two, and therefore cannot be used to test the hypothesis of Rayner and O'Brien (2001) that

inversions of column data are less sensitive to transport model errors than inversions of in situ data. To test this hypothesis, we devised two purely theoretical in situ networks, called "IS-LNLG" and "IS-OG". The IS-LNLG (IS-OG) network consists of PBL samples of the $CO_2$ mole fraction at locations and times of all OCO-2 land (ocean) soundings from § 2.3.2. The five different model fields were sampled at the IS-LNLG and IS-OG networks, 30 m above ground level as defined by the 1 arc-minute global relief model ETOPO01 (Amante and Eakins, 2009). The difference in fluxes between inversions of IS-LNLG

(IS-OG) in situ pseudo-data and LNLG (OG) OCO-2 pseudo-data can be expected to reflect the difference between PBL and column sampling over land (ocean), and not differences in spatiotemporal coverage between actual in situ and OCO-2 samples.

### 2.4 Inversion framework

TM5 4DVAR is a state-of-the-art variational inversion system that has been used to estimate surface fluxes of $CO_2$ (Basu et al., 2013), CO (Krol et al., 2013), $CH_4$ (Bergamaschi et al., 2013) and $N_2O$ (Corazza et al., 2011). Given a set of prior fluxes $\boldsymbol{x}_a$

with their error covariance $S_a$, a set of measurements $\boldsymbol{y}$ with their error covariance $S_\epsilon$, and a transport model $K$ connecting fluxes to measurements, a Bayesian flux estimation system tries to minimize the cost function $J$

$$J = \frac{1}{2}(K\boldsymbol{x} - \boldsymbol{y})^T S_\epsilon^{-1}(K\boldsymbol{x} - y) + \frac{1}{2}(\boldsymbol{x} - \boldsymbol{x}_a)^T S_a^{-1}(\boldsymbol{x} - \boldsymbol{x}_a) \tag{1}$$

The posterior estimate of $\boldsymbol{x}$, usually denoted $\hat{\boldsymbol{x}}$, is given by (Rodgers, 2000)

$$\hat{\boldsymbol{x}} = \boldsymbol{x}_a + S_a K^T \left(K S_a K^T + S_\epsilon\right)^{-1}(\boldsymbol{y} - K\boldsymbol{x}_a) = \boldsymbol{x}_a + G(\boldsymbol{y} - K\boldsymbol{x}_a) \tag{2}$$

where $G = S_a K^T \left(K S_a K^T + S_\epsilon\right)^{-1}$ is called the Kalman gain matrix and determines the weighting between prior information and observations. Details about TM5 4DVAR have been documented by Meirink et al. (2008). In this work we use the ability of TM5 4DVAR to assimilate in situ and total column $CO_2$ measurements as documented by Basu et al. (2013). We run the TM5 transport model ($K$ in the equation above) at global 3°×2°×25 layer resolution, and solve for ocean and land fluxes at 3°×2° globally. We have already described our method for constructing the synthetic observations $\boldsymbol{y}$. Below we describe the

remaining elements of this inversion, namely $S_a$, $S_\epsilon$ and $\boldsymbol{x}_a$.

### 2.4.1 Prior flux ($x_a$) and covariance ($S_a$)

Prior ocean and land fluxes were constructed as the multi-year (2000-2015) mean of CarbonTracker 2016 posterior fluxes (https://www.esrl.noaa.gov/gmd/ccgg/carbontracker/). Hence, the prior did not have any interannual variability, but did have a land sink consistent with the decadal trend of atmospheric $CO_2$ growth rate. Fossil fuel emissions, for both the true and prior fluxes, were taken from the ODIAC inventory (Oda and Maksyutov, 2011) and not optimized. Both the land and ocean fluxes were optimized on a weekly time scale, on a global 3°×2° grid. Ocean and land fluxes had 3-hourly variations within each week, which were not optimized. The fossil fuel flux had daily and hourly variations according to Nassar et al. (2013). Errors in the weekly prior ocean fluxes were assumed to be 1.57 times the absolute flux in each grid cell, with a spatial correlation of 1000 km and a temporal correlation of 3 weeks. Errors in the weekly prior terrestrial fluxes were assumed to be half the heterotrophic respiration in each grid cell from the CASA biosphere model (Randerson et al., 1996), with a spatial correlation of 250 km and a temporal correlation of 1 week. The grid scale uncertainty on terrestrial fluxes thus constructed was typically an order of magnitude higher than for ocean fluxes. However, due to the shorter error correlation lengths and times assumed for terrestrial fluxes, the uncertainties on the global totals for 2015 were of the same order of magnitude, 0.44 PgC/yr for oceans and 0.53 PgC/yr for land. The ocean uncertainty constructed this way corresponds roughly to the uncertainty on the ocean sink imposed by decadal measurements of the atmospheric $O_2/N_2$ ratio (Keeling and Manning, 2014), while the land flux uncertainty is large enough to allow sufficient summertime uptake over North America and Eurasia (Basu et al., 2016).

### 2.4.2 Data error ($S_\epsilon$)

The analytical error of a flask-air or continuous in situ measurement of $CO_2$ is very small, typically 0.1-0.2 ppm. However, even with perfect fluxes and an unbiased transport model, we do not expect to fit all observations to that precision, because a coarse resolution transport model cannot adequately represent sub-grid scale variations that lead to the measured mole fraction at a point. Therefore $S_\epsilon$ also contains the representativeness error of the transport model, which can be considered to be a random error contributed by the model. This representativeness error is computed by evaluating the norm of the spatial gradient of the modeled $CO_2$ mole fraction at the scale of TM5's lateral resolution at each sampling time and location. The total error in $S_\epsilon$ is the quadrature sum of this model error and an analytical error of 0.2 ppm. Figure 2 shows the total and analytical errors at three example sites at times when $CO_2$ samples were taken. Tutuila, American Samoa (SMO) is a remote marine boundary layer site with little model variability, with a model error of ~1 ppm. Niwot Ridge (NWR) is a background mountaintop site with the continental US, and therefore has higher model variability. Finally, Beech Island (SCT) is a tall tower in the southeastern US where seasonally coherent transport variability is convolved with strong local fluxes. It should be noted here that the numbers in figure 2 are somewhat smaller than typical values in the literature (e.g., Baker et al., 2006; Peylin et al., 2013). Therefore, our estimate of the transport uncertainty for in situ $CO_2$ inversions is likely to be on the higher side.

The formal reported uncertainty of OCO-2 $XCO_2$ retrievals is an underestimate (Worden et al., 2016). Therefore, the errors estimated for the 10 s averages are likely underestimates as well. Moreover, $S_\epsilon$ in equation (1) is not just the measurement

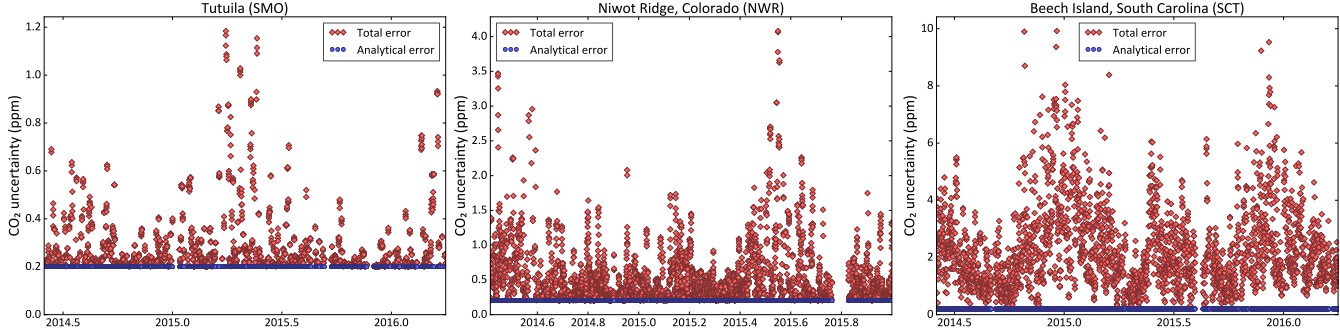

**Figure 2.** Analytical (blue) and total (red) uncertainty on in situ measurements in the $S_\epsilon$ matrix at three example sites, at times of actual $CO_2$ measurements. SMO is a remote, marine boundary layer site with little model variability, while LEF and WKT are continental sites with significant model variability.

error, but the covariance of the model-observation mismatch. Therefore, we construct the data error for $XCO_2$ as the sum of two components, $\sigma_{10s}^2 = \sigma_{meas}^2 + \sigma_{model}^2$.

The measurement part, $\sigma_{meas}^2$, is calculated in two steps. First, variances are calculated for 1 s averages by summing the inverse variances of all the soundings in that average, as reported by the retrieval algorithm. A lower threshold of $\varepsilon_{base}^2/N_{ret}$ is

set on that variance, where $N_{ret}$ is the number of retrievals in the 1 s average, and $\varepsilon_{base}$ is an error floor that is 0.8 ppm over land and 0.5 ppm over oceans. If the 1 s variance calculated this way is denoted $\sigma_{1s}^2$, then the variance on the 10 s average is calculated as $\sigma_{meas}^{-2} = (1/10) \sum \sigma_{1s}^{-2}$, where the sum goes over the 1 s bins in the 10 s average. Note that the final error $\sigma_{meas}$ does not drop by $\sqrt{10}$ because of the factor $1/10$ in the front.

The model part, $\sigma_{model}$, is calculated by considering a suite of inverse models optimized against in situ data, and calculating

their difference with OCO-2 $XCO_2$ retrievals. The differences are binned by latitude band, month and OCO-2 sounding mode, and averaged. For each month/latitude/mode bin, the cross-model spread in the average differences is taken to be $2 \times \sigma_{model}$ for that bin. While there is no unique way of deriving a $\sigma_{model}$, this algorithm creates a $\sigma_{model}$ that includes model variability across multiple state-of-the-art transport models driven by realistic fluxes. In practice, $\sigma_{model}$ is usually larger than $\sigma_{meas}$ for most 10 s averages. On average, $\sigma_{10s}$ is $\sim$1.5 ppm and $\sim$0.9 ppm for land and ocean soundings respectively.

One final point to note is that in OSSEs, random perturbations are often added to the data to simulate random measurement error (e.g., Chevallier et al., 2010). However, that is relevant when the goal is to get an accurate estimate of the analytical posterior uncertainty of the flux. In this work, however, the goal is to estimate the spread in flux estimates due to the relative bias between different transport models. Moreover, inversion groups assimilating real OCO-2 and surface data do not add random error to those measurements, so differences in flux estimates between different groups have no contribution from

this kind of added random measurement error. Therefore, in this work we have not added any perturbations to our synthetic measurements.

### 2.4.3 Note about the impact of transport models

If two different transport models ($K_1$ and $K_2$) are used to assimilate data $\boldsymbol{y}$ starting from the same prior $\boldsymbol{x}_a$ and with the same error matrices $S_a$ and $S_\epsilon$, then their respective posterior flux estimates will be (Rodgers, 2000)

$$\hat{\boldsymbol{x}}_i = \boldsymbol{x}_a + \left( I - \hat{S}_i S_a^{-1} \right)(\boldsymbol{x}_t - \boldsymbol{x}_a) \tag{3}$$

$$\hat{S}_i = \left( S_a^{-1} + K_i^T S_\epsilon^{-1} K_i \right)^{-1} \tag{4}$$

Where $\boldsymbol{x}_t$ is the true flux. Therefore the difference between the two flux estimates will be

$$\hat{\boldsymbol{x}}_1 - \hat{\boldsymbol{x}}_2 = \left( \hat{S}_2 - \hat{S}_1 \right) S_a^{-1}(\boldsymbol{x}_t - \boldsymbol{x}_a) \tag{5}$$

That is, the transport related flux difference depends on the distance from the prior to the true flux, as well as $\hat{S}_i$, which is determined by the interaction between the error matrices and the transport model $K_i$. However, equation (5) makes a crucial assumption, namely that both transport models are unbiased, or $\boldsymbol{y} = K_i \boldsymbol{x}_t + \epsilon$, where $\epsilon$ is the random error of $\boldsymbol{y}$. In practice, this is never the case, and for flux inversions the error due to a transport model is usually because the transport model is biased with respect to true atmospheric transport, at spatiotemporal scales of interest. In our experiment, we mimic this by letting "nature" be each of five transport models (TM5, PCTM, LMDZ, ACTM, GEOS-Chem) in turn. As long as these models span the range of transport in nature (Patra et al., 2011), the uncertainty in fluxes coming out of our experiment will be a reasonable estimate of the uncertainty due to the difference between modeled and true atmospheric transport. In our experiment, the difference between two flux estimates from pseudo-data produced by two different transport models $K_1$ and $K_2$ is

$$\hat{\boldsymbol{x}}_1 - \hat{\boldsymbol{x}}_2 = \hat{S} K^T S_\epsilon^{-1}(K_1 - K_2)\boldsymbol{x}_t \tag{6}$$

where $\boldsymbol{x}_t$ are the true fluxes in our OSSE, and $\hat{\boldsymbol{x}}_i$ is the flux estimate when synthetic observations produced by model $K_i$ are assimilated in TM5 4DVAR. $K$ represents the transport and observation operator of TM5, while $\hat{S}$ depends on $K$, $S_a$ and $S_\epsilon$. In a real data inversion, flux estimates from two different inversion frameworks that happen to use transport models $K_1$ and $K_2$ will not necessarily differ by the amount given in equation (6), because of other choices made in setting up the inversion systems. Rather, equation (6) can be thought of as the range of flux estimates possible in a typical flux inversion (TM5 4DVAR in our case) if $K_1$ and $K_2$ span the range of possible real atmospheric transport. It should be noted that the range as expressed in equation (6) does not depend on the flux prior $\boldsymbol{x}_a$, but only on the prior uncertainty $S_a$ through its influence on $\hat{S}$.

## 2.5 Difference between transport models

OCO-2 has a local overpass time of 1:30 PM, and most surface measurements assimilated in flux inversions – except for mountaintop sites – are from the afternoon once a fully mixed planetary boundary layer (PBL) has formed. Therefore, the mid-afternoon $CO_2$ mole fraction difference between models, both in the PBL and in the total column, would contribute to flux differences in our experiment. The zonal average of those differences between Dec 1 2014 and Mar 1 2016 are plotted in figure 3, where the lowest $150\,\text{hPa}$ is an approximation for the mid-afternoon PBL depth. Maps of these differences for

summer, winter and the annual average are shown in figures B1 and B2 in the appendix. For each grid cell, the median $CO_2$ mole fraction of all five models was subtracted from each model to highlight model differences instead of large scale features common to all models. All modeled $CO_2$ fields were mapped to a global $1° \times 1°$ grid while conserving mass. Since the models had varying resolutions and grid registrations, this resulted in unavoidable checkered patterns in the differences in figure 3. That, however, did not impact the large scale model to model differences shown.

In figure 3, the agreement across models is generally better over the Southern Hemisphere (SH) than over the north. This is primarily driven by larger ocean masses in the south than in the north, since as figures B1 and B2 show, the agreement across models is generally higher over oceans than over land. This is expected because (a) vertical transport, one of the major axes of variability across models, is stronger over land than over oceans, and (b) surface flux variability is also higher over land than over oceans, amplifying the difference between transport models when viewed in the $CO_2$ concentration space. Models driven by the same parent meteorology do not necessarily show the same features in the modeled $CO_2$ field. In the Northern Hemisphere (NH) summer, LMDZ shows faster exchange between the continental PBL and the free troposphere (FT) than TM5, evidenced by higher $CO_2$ mole fractions in the continental PBL in figure B1. By similar logic, PCTM shows much slower PBL-FT exchange than GEOS-CHEM. In the NH winter, contrary to summertime, in the northern temperate latitudes PCTM and TM5 exhibit faster PBL-FT exchange compared to GEOS-CHEM and LMDZ respectively. The two models driven by GEOS-derived winds (GEOS-CHEM and PCTM) are significantly different in the PBL over North and South America, East Asia and Tropical Africa throughout the year. The corresponding difference between the two models driven by ERA Interim winds (LMDZ and TM5) are smaller. ACTM has an overall low bias of $\sim 0.5\,$ppm in the PBL, which shows up to a lesser extent in the total column (figure 3) and the total atmospheric $CO_2$ mass (figure 1). However, such an overall bias should not affect fluxes estimated from ACTM pseudo-observations. ACTM also appears to trap more (compared to the model median) of the wintertime respiration signal from Boreal Eurasia in the PBL (figure B1), which should have implications for Boreal flux estimates.

In the total column, GEOS-CHEM and PCTM look very different in the NH summer, with PCTM trapping more of the NH summertime uptake and SH wintertime respiration signals in the respective hemispheres. In the NH winter, GEOS-CHEM displays the Tropical Asian biomass burning signal more strongly in the total column than PCTM, while the East Asian fossil fuel enhancement is higher in the GEOS-CHEM $XCO_2$ throughout the year. In the NH summer, LMDZ appears to transport more of the Temperate and Boreal uptake signal to the south compared to TM5, leading to slightly higher $XCO_2$ values in the north. In the NH winter, conversely, TM5 appears to transport more of the northern respiration signal to the south.

## 3   Results

Figure 4 shows the range of the annual $CO_2$ flux from assimilating synthetic observations produced by the five different transport models. For each region, the black horizontal line denotes the estimate from assimilating pseudo-obs generated by TM5, i.e., it is the "perfect transport" OSSE. The other four models are not distinguished here for visual clarity, but figure D1 in Appendix D marks them separately. The range of the annual flux estimates across the five forward models in figure 4, which

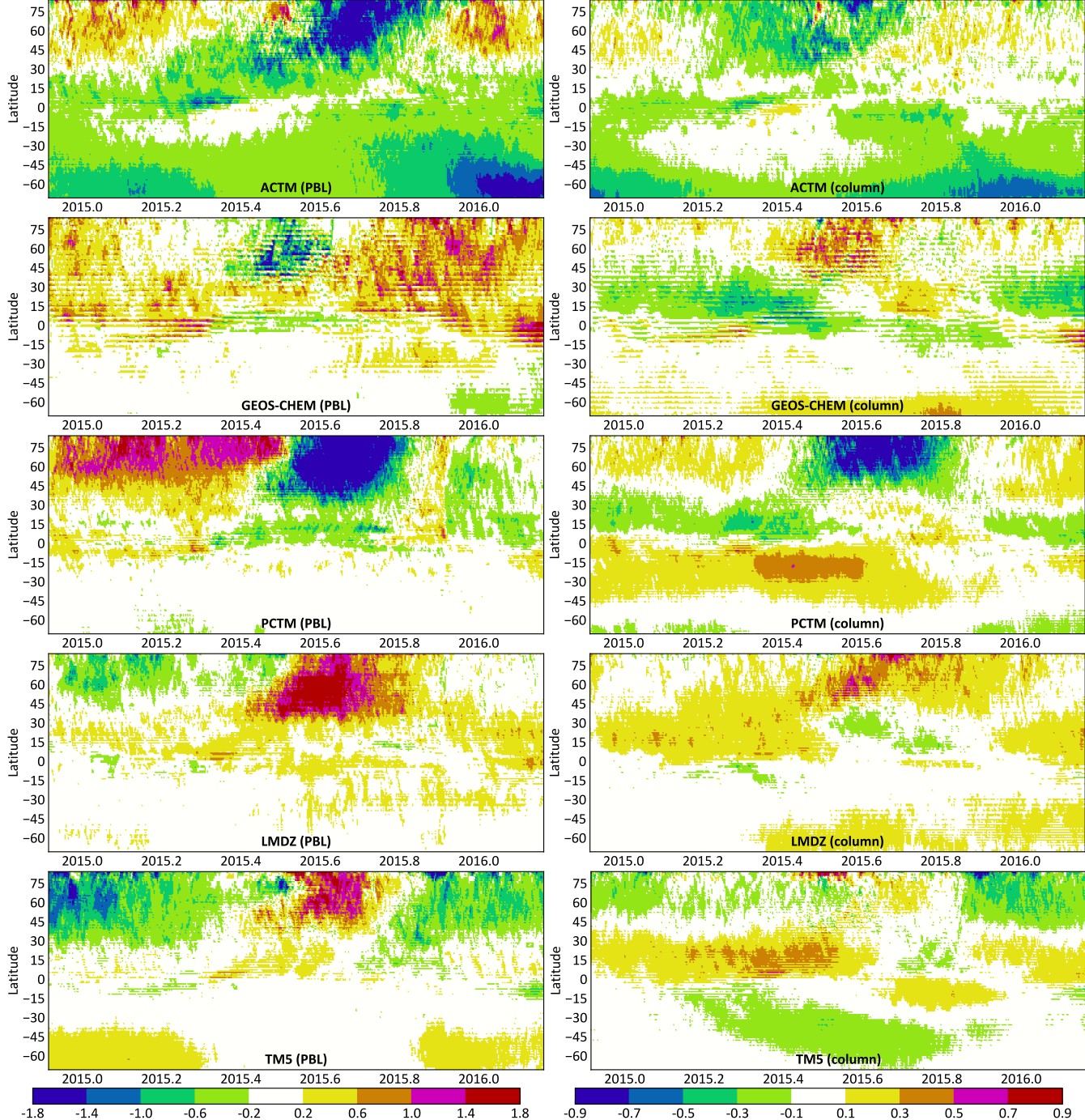

**Figure 3.** The zonal average difference between each model (ACTM, GEOS Chem, LMDZ, PCTM and TM5) and the cross-model median at 1:30 PM local time, in ppm $CO_2$, between Dec 1 2014 (2014.915) and Mar 1 2016 (2016.164). The left column depicts differences in the lowest 150 hPa, which is an approximation for the PBL. The right column depicts differences in column averaged $CO_2$. Each column has its own color bar. Since transport differences in the total column are smaller than in the PBL, the dynamic range of the right column is half that of the left column.

is a measure of the transport model uncertainty in the flux estimates, is tabulated for all regions and data streams in table C1 in the appendix.

Real satellite retrievals of $XCO_2$ have spatially coherent and sampling mode-dependent biases due to interfering species such as aerosols and water, surface effects such as albedo and elevation, and geometric effects such as the solar zenith angle. However, synthetic data generated by the five transport models, which serve as the input in our inversions, do not have such biases. Hence the range of flux estimates from different data sets is purely determined by the coverage difference between different sampling modes and the type of measurement (total column versus near-surface point), while the differences between the flux estimates from pseudo-obs generated by different models (horizontal lines within each colored bar in figure 4) is a measure of the inter-model transport difference as sampled by a particular observing mode/network. In this context, the horizontal black lines in figure 4 represent "perfect transport" inversions, meaning the synthetic observations were generated and assimilated with the same transport model. Therefore, the difference between those lines (TM5) and true fluxes (white circles) in the figure represents the balance between $S_a$ and $S_\epsilon$ in our setup of TM5 4DVAR, and a smaller difference from a different model (any other horizontal line) should not be interpreted as significant. It should also be noted that our goal is not to rank models according to their proximity to true fluxes in figures 4 and D1, but rather to quantify the spread across different models used to generate the synthetic data, and how that spread varies with sampling and coverage.

Figures 5 and 6 show the range of monthly fluxes from TRANSCOM-like land and ocean regions for each type of synthetic data stream assimilated. For visual clarity, only the range across the five models has been shown instead of individual flux estimates. The land regions in figure 5 are identical to the TRANSCOM regions, except that Africa has been partitioned into Saharan and sub-Saharan Africa instead of north and south of the equator.

## 4 Discussion

### 4.1 Global budget

All five models were run from the same initial $CO_2$ field with the same surface fluxes. The resulting global burden of $CO_2$ in the models were close but slightly different, as shown in figure 1. The increase in the global average $CO_2$ mole fraction between Jan 1 2015 and Jan 1 2016 ranged from 2.89 ppm (TM5) to 2.97 ppm (LMDZ). That 0.08 ppm range in the mole fraction, given the dry air mass of TM5, corresponds to a range of 0.16 PgC in the change in the global $CO_2$ burden over 2015. Therefore, even if our pseudo-data inversions nail the global $CO_2$ budget for 2015 exactly, we can expect a variation of up to 0.16 PgC in that budget owing to the small model-to-model differences in figure 1.

The global total $CO_2$ flux in figure 4 shows a spread of $\sim$1.5 PgC/yr for in situ inversions, which is larger than the spread seen in earlier inverse model intercomparisons such as Peylin et al. (2013). This is because intercomparisons such as Peylin et al. (2013) typically report the constraint on the multi-year average global growth rate, while here we are looking at the constraint on a single year's growth rate from in situ samples. Houweling et al. (2015) compared eight different inverse models of a single year using in situ data, and found a spread of 1.73 PgC/yr across models for the annual growth rate, with a standard deviation of 0.5 PgC/yr. The inversions in Houweling et al. (2015) were less controlled compared to our setup, since they used different

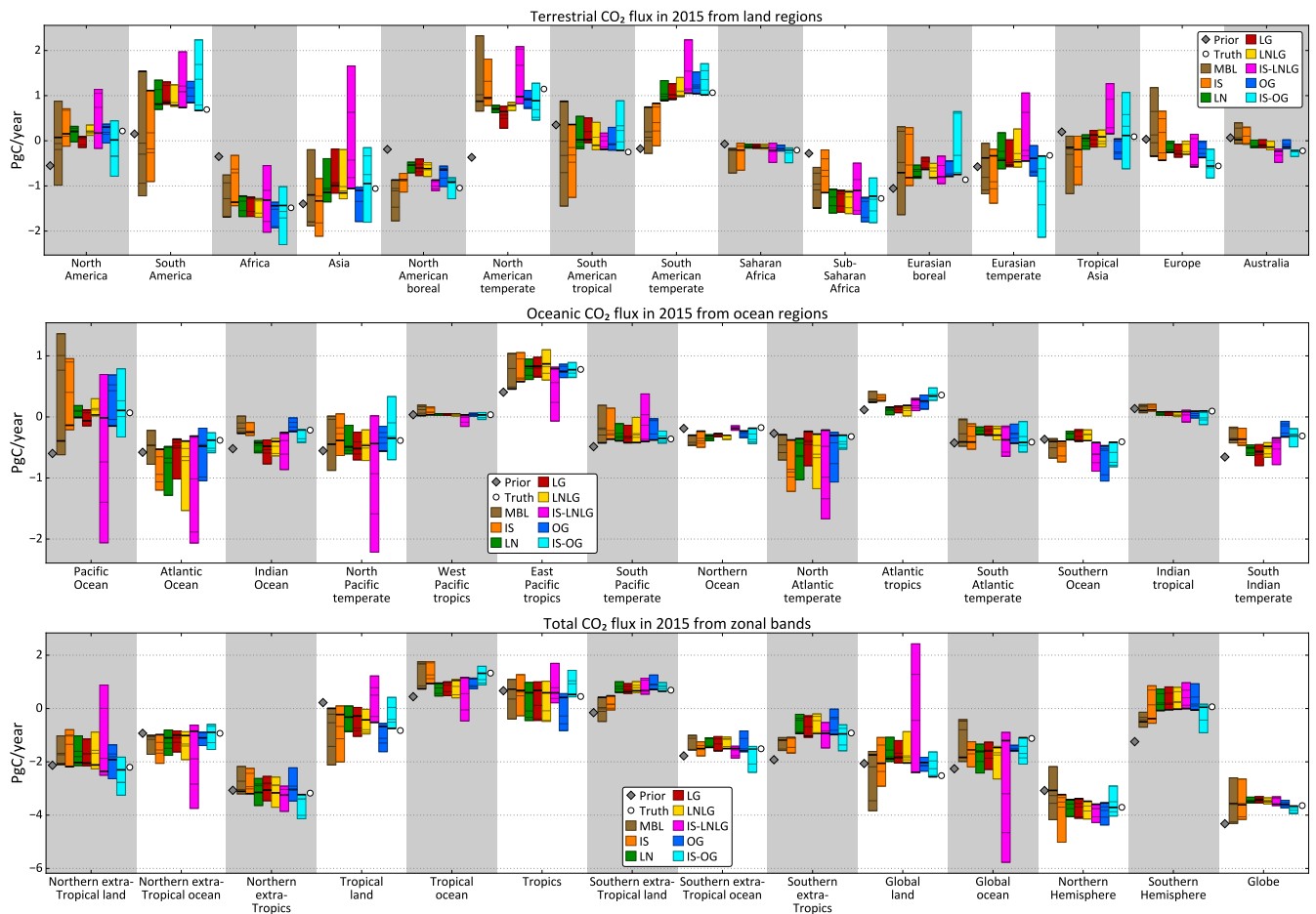

**Figure 4.** Annual flux estimates from land (top) and ocean (middle) regions and zonal bands (bottom). For each region, the prior and true fluxes are shown by a grey diamond and a white circle respectively. The different colored bars correspond to different synthetic data streams assimilated; IS = in situ, LN/LG/OG = OCO-2 land nadir/land glint/ocean glint, and LNLG = LN + LG (all land soundings). The data streams IS-LNLG and IS-OG are theoretical PBL sampling networks at $OCO_2$ sounding locations and times, described in § 2.3.3. For each color, the vertical extent of the bar denotes the range (minimum to maximum) of the flux estimates from pseudo-data produced by the five transport models for that data stream. The black horizontal line through each bar denotes the estimate from TM5 pseudo-obs, while the fainter horizontal lines denote the estimates from the pseudo-obs produced by the other four models. The individual models are not distinguished here for visual clarity, but are marked separately in figure D1 in Appendix D.

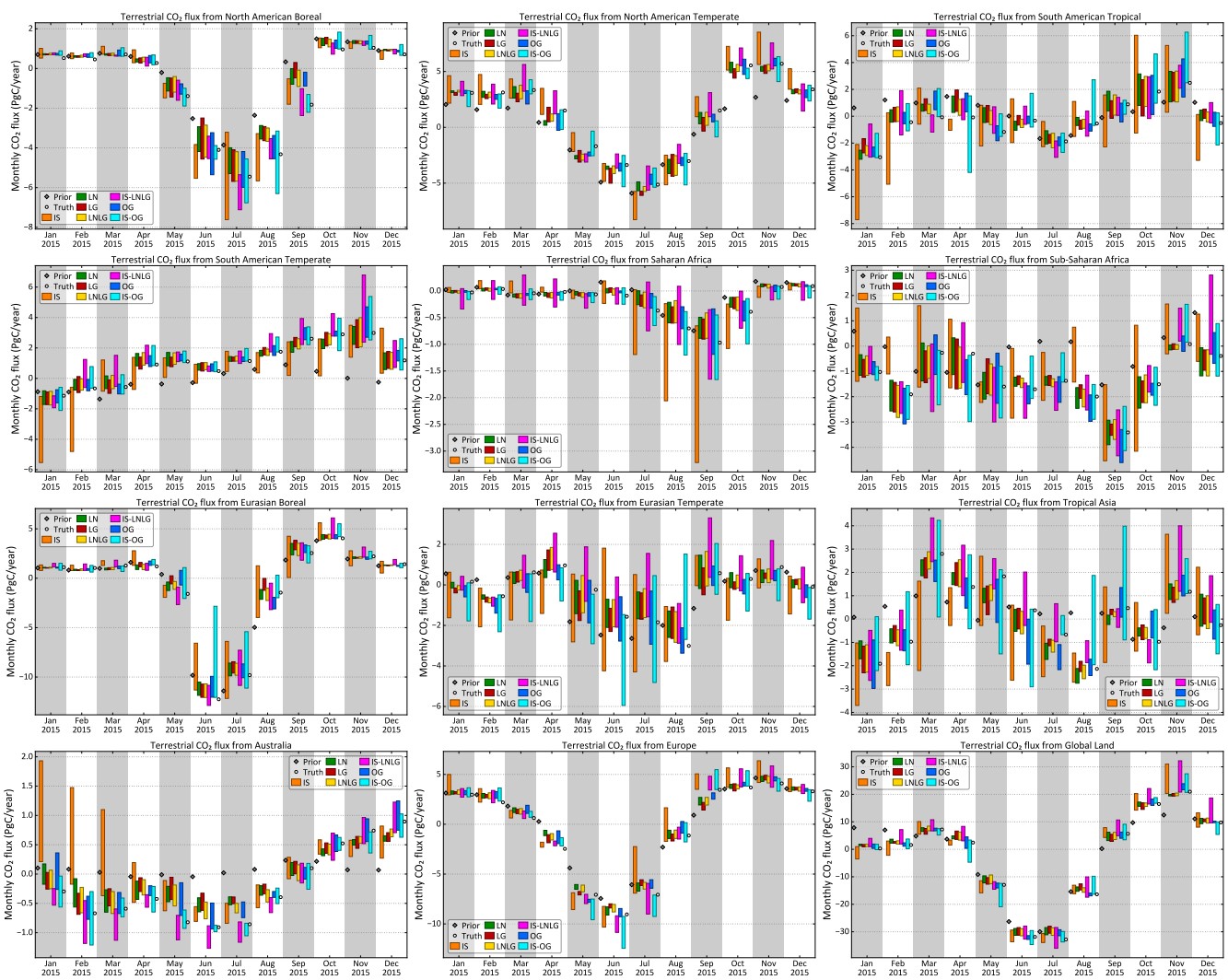

**Figure 5.** Monthly flux estimates from TRANSCOM-like land regions and global total land. The different colors correspond to different synthetic data streams assimilated, as in figure 4. The different models used to generate the synthetic data have not been distinguished here to minimise visual clutter. The theoretical PBL networks IS-LNLG and IS-OG have also been omitted for the same reason. Plots of seasonal fluxes over many more regions, with the models distinguished, are included in the supplementary material.

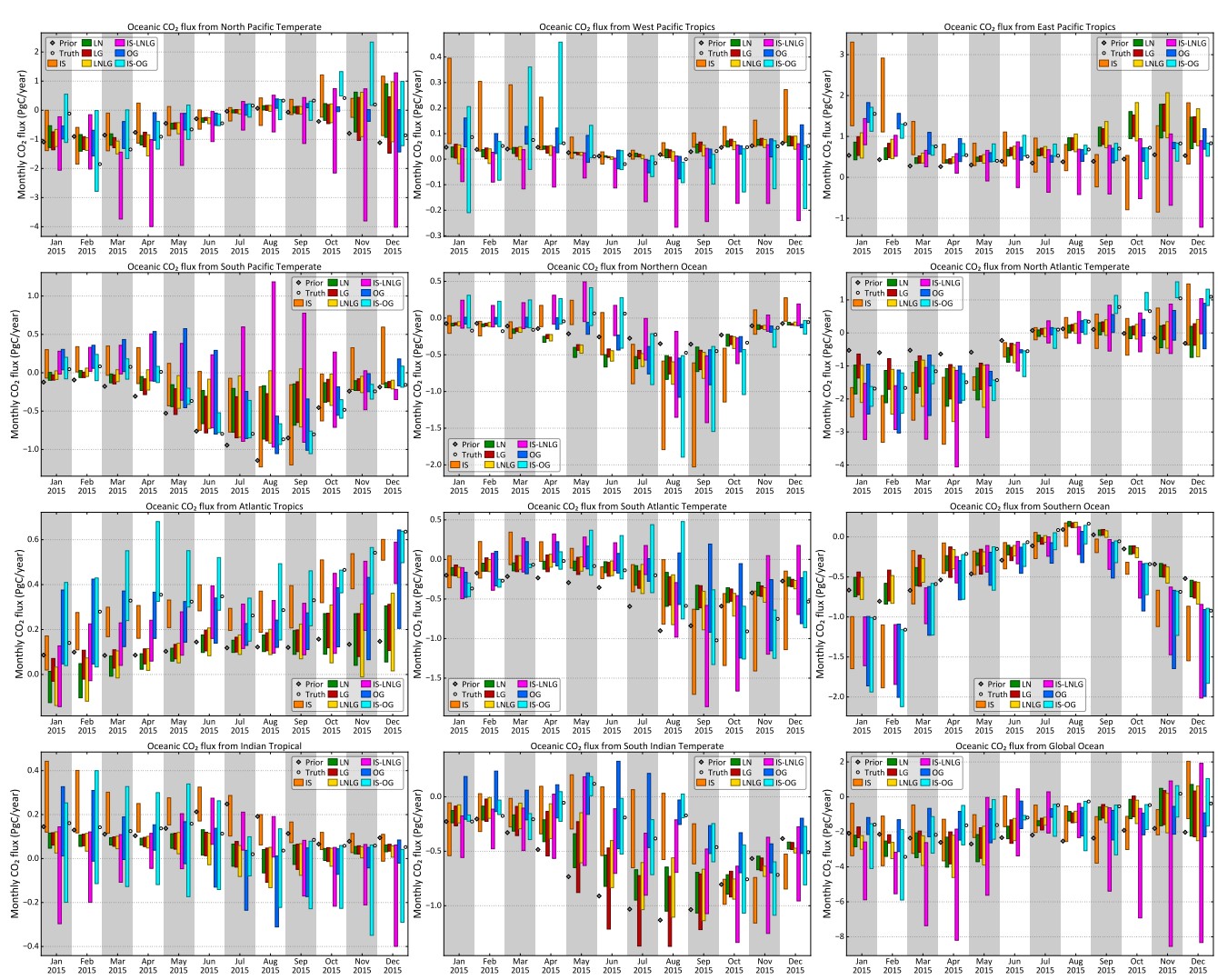

**Figure 6.** Same as figure 5, except over TRANSCOM ocean regions and global total ocean.

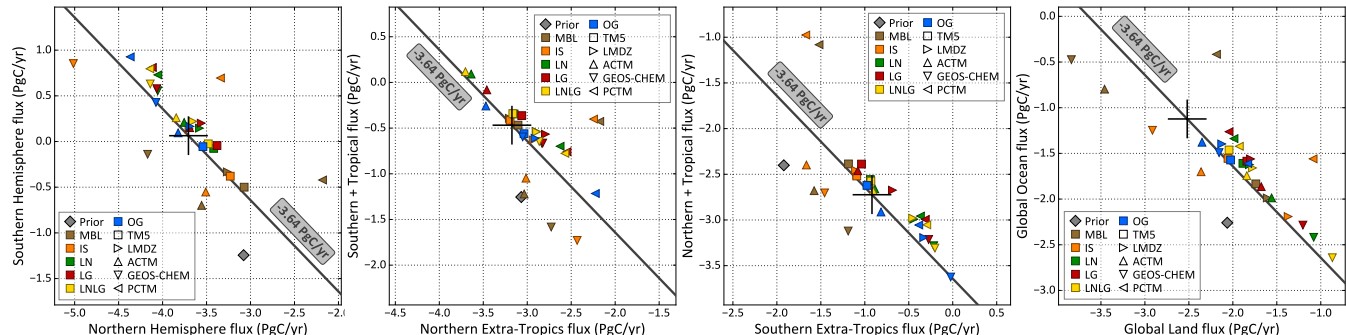

**Figure 7.** The partitioning of the 2015 global $CO_2$ sink into two geographical domains, with the tropics being defined as 23.5°north and south latitudes. Each color represents one type of synthetic data assimilated, while each symbol shape represents one model used to generate the synthetic data. The diagonal gray line represents the 2015 global sink of 3.64 PgC/yr in the true fluxes used to generate the synthetic data, while the large plus sign denotes their partitioning. The scales are identical across all four figures, but not the origins.

flux and measurement covariances as well as different transport models. Therefore, in our more controlled experiment, a spread of 1.5 PgC/yr is reasonable among the different in situ data streams. It is noteworthy that the spread in the global total flux in figure 4 for the OCO-2 pseudo-data inversions is ∼0.25 PgC/yr, close to the previously calculated limit of 0.16 PgC/yr. This reduction in the spread from in situ to OCO-2 inversions is primarily due to the more spatially extensive sampling of OCO-2 and not because of OCO-2's sensitivity to the total column (as opposed to the surface layer), evidenced by the ∼0.25 PgC/yr spread in the global $CO_2$ flux from IS-LNLG and IS-OG inversions in figure 4. This suggests that compared to the current in situ network, a more spatially extensive sampling strategy, whether total column or PBL, can provide a stricter constraint on the global $CO_2$ budget that is less sensitive to transport model specifics.

### 4.2 Large scale partitioning of the global budget

**Table 3.** The spread in the flux partitioning across five models from the assimilation of different pseudo-data streams. This is a tabulated summary of the information in figures 7 and 8. For each pseudo-data stream (e.g., MBL) and each partitioning (e.g., 23.5 °N, which is the dividing line between the northern extra-tropics and the rest), the table contains the spread across five models of the sum and difference of the fluxes between the two partitions. All numbers are in PgC/year.

| Partitioning | MBL | | IS | | LN | | LG | | LNLG | | OG | | IS-LNLG | | IS-OG | |
|---|---|---|---|---|---|---|---|---|---|---|---|---|---|---|---|---|
| | sum | diff | sum | diff | sum | diff | sum | diff | sum | diff | sum | diff | sum | diff | sum | diff |
| Equator | 1.71 | 2.27 | 1.51 | 3.02 | 0.22 | 1.44 | 0.24 | 1.59 | 0.24 | 1.49 | 0.29 | 1.81 | 0.33 | 1.66 | 0.29 | 2.13 |
| Land/ocean | 1.71 | 3.74 | 1.51 | 2.49 | 0.22 | 1.99 | 0.24 | 1.86 | 0.24 | 2.35 | 0.29 | 0.75 | 0.33 | 9.71 | 0.29 | 1.92 |
| 23.5 °N | 1.71 | 1.67 | 1.51 | 2.08 | 0.22 | 1.80 | 0.24 | 1.59 | 0.24 | 2.03 | 0.29 | 2.20 | 0.33 | 1.62 | 0.29 | 1.65 |
| 23.5 °S | 1.71 | 2.37 | 1.51 | 2.12 | 0.22 | 1.44 | 0.24 | 1.59 | 0.24 | 1.46 | 0.29 | 1.95 | 0.33 | 1.61 | 0.29 | 1.84 |

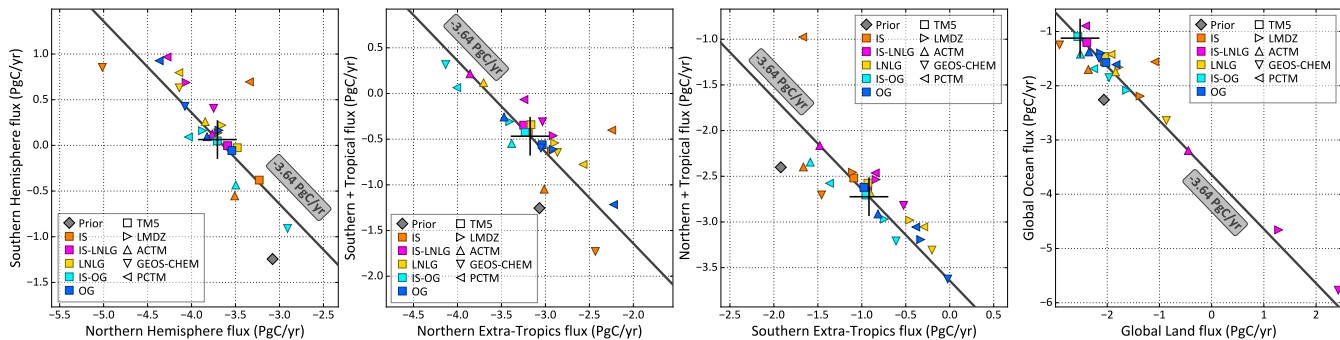

**Figure 8.** The partitioning of the 2015 global $CO_2$ sink into two geographical domains, with the tropics being defined as 23.5°north and south latitudes. This is similar to figure 7, except that we have compared two real OCO-2 and one real in situ sampling schemes (LNLG, OG, IS) with the two theoretical in situ ones (IS-LNLG, IS-OG) of § 2.3.3. The scales are identical across three of the four figures and the same as figure 7; the partitioning of global and ocean fluxes had significantly more spread, and required a different scale.

The global atmospheric growth rate of $CO_2$ (denoted C below) is determined by the fossil fuel ($F_{ff}$) emissions and the global sink from the land biosphere ($F_{bio}$) and oceans ($F_{oce}$)

$$\frac{dC}{dt} = F_{ff} + F_{bio} + F_{oce} \tag{7}$$

where $F_{bio}$ includes fire emissions. $CO_2$ inversions typically assume a known $F_{ff}$ and estimate $F_{bio}$ and $F_{oce}$ from atmospheric
observations of $CO_2$. Therefore, in a suite of inversions assuming the same $F_{ff}$, the global total sink $F_{bio} + F_{oce}$ is constrained
to a number whose uncertainty is determined by how well the global $CO_2$ budget is determined by the $CO_2$ observations
assimilated. A plot of the estimated $F_{oce}$ versus $F_{bio}$ from the suite should therefore be clustered around a straight line with a
slope of $-1$. The same logic applies for any other two-way partitioning of the global sink, such as northern versus southern
hemisphere. Figures 7 and 8 show four different two-way partitionings of the global total $CO_2$ sink from our ensemble of
inversions of synthetic data. The straight line with slope $-1$ corresponds to the global total sink of -3.64 PgC/year in our true
fluxes used to generate the observations. For each inversion estimate, the distance from that straight line is a measure of how
much the estimated global budget deviates from the true global budget for 2015, while the position along the line is an indication
of how the inversion splits the global budget into the two partitions. Table 3 contains summary statistics from figures 7 and 8.
For each data stream (e.g., MBL) and partitioning (e.g., Equator, which corresponds to the partitioning between the northern
and southern hemispheres), the table contains the spread in the sum and difference of fluxes between the two partitions. The
spread in the sum is a measure of the uncertainty in the global budget as constrained by that data stream, while the spread in
the difference is indicative of the uncertainty in the partitioning.

The global budget for a single year is constrained poorly by inversions with IS and MBL pseudo-data, evidenced by the
large spread of the global sum in table 3 and the scatter of the IS and MBL points around the -3.64 PgC/yr straight line in
figures 7 and 8. This is consistent with the larger spread in the global sink estimate of inversions with IS and MBL data in
figure 4. Among the models, PCTM pseudo-obs seem to demand a higher $CO_2$ flux consistently, while ACTM and GEOS-

CHEM pseudo-obs demand slightly lower $CO_2$ fluxes. Since growth in the atmospheric $CO_2$ burden was the same for all the models in 2015 (figure 1), these differences are due to large scale transport differences sampled by the in situ network.

Since the OCO-2 pseudo-obs in this OSSE are bias free, differences in the partitioning from different sounding modes (LN, LG, OG and land or LNLG) are purely due to sampling differences. This includes the obvious difference of sampling the atmosphere over land and ocean surfaces, and also a more subtle difference in the timing of the samples, coming from the fact that during the early part of the OCO-2 record up to July 2015, the satellite operated continuously for 16 days in nadir (glint) mode before switching to glint (nadir). As a result, land nadir and land glint samples over the same location could be separated by up to 16 days. Since $CO_2$ fluxes can change significantly over 16 days, this may give rise to differences in LN and LG derived flux estimates. The impact of spatiotemporal differences in sampling are evident in figure 7. Among assimilations of OCO-2 pseudo-obs (LN, LG, LNLG, OG) simulated by a single forward model, there can be a ~0.5 PgC/yr spread in the partitioning across a latitude, whether the equator or one of the tropics, while the land-ocean partitioning is more uncertain, with a spread of up to ~1.5 PgC/yr. Interestingly, the land-ocean partitioning seems to be better pinned down by OCO-2 ocean soundings than land soundings, evidenced by the smaller inter-model spread when assimilating OG pseudo-obs than when assimilating LN, LG or LNLG pseudo-obs. The same does not appear to hold for any latitudinal partitioning.

Finally, we contrast the partitioning from LNLG (OG) with that from IS-LNLG (IS-OG) to gauge the impact of transport error on PBL versus total column measurements. The IS-LNLG (IS-OG) network, which has spatially extensive PBL sampling only over land (ocean), has a much larger spread in the land/ocean partitioning compared to the LNLG (OG) network of column samples. This suggests that if the goal is to partition land and ocean fluxes, PBL sampling can amplify differences across transport models, which are larger in the PBL than in the total column (Rayner and O'Brien, 2001). Moreover, comparing the spreads of IS-LNLG and IS-OG inversions suggests that these transport differences are larger over land than over ocean. If the goal, however, is to partition the global budget across a latitude (i.e., the other three partitionings in table 3), column sampling does not appear to have an obvious advantage over PBL sampling. This is likely because of the fast zonal mixing of the $CO_2$ flux signal, i.e., the flux signal missed by PBL samples at one location due to incorrectly modeled vertical mixing will be seen by PBL sites downstream within the same zonal band.

## 4.3 Annual fluxes at zonal, continental and TRANSCOM scales

The spread in flux estimates across the five forward models, or the transport-driven uncertainty, is very similar in figure 4 and table C1 between IS and MBL data streams for most regions. Over some land regions that have seen a significant increase in measurement density since Baker et al. (2006), such as North America and Europe, the additional measurements in IS result in a smaller uncertainty compared to MBL. Over land regions where the coverage of IS and MBL are almost identical, such as Africa and Tropical Asia, the uncertainties are (not surprisingly) comparable between IS and MBL. Over ocean regions, the IS and MBL uncertainties are very similar, except over the Pacific, where the increased coverage in IS on the west coast of North America is likely responsible for the reduction in uncertainty. The uncertainty in the global uptake and the global land and ocean fluxes are slightly smaller for the IS network compared to the MBL network. However, for most other zonal regions the IS and MBL uncertainties are roughly equal, likely because of the fast zonal mixing in the atmosphere.

The regional annual flux estimates of figure 4 show that the spread among land flux estimates when assimilating OCO-2 pseudo-data over land (LN, LG and LNLG) is often smaller than when assimilating in situ data (IS, MBL). This could be a combination of the total column nature of OCO-2 pseudo-data and its increased spatial homogeneity of coverage. To separate the two effects, we look at IS-LNLG, which has the same coverage as LNLG but only PBL samples instead of total columns.

Over certain regions, such as temperate North and South America, and temperate Eurasia, the IS-LNLG spread is larger than the IS spread, which is larger than the LNLG spread. This suggests that over those regions, the transport model error – relative to the flux signal – in the total column is smaller than in the PBL, leading to lower transport-drive uncertainty in total column $CO_2$ assimilations than in situ $CO_2$ assimilation. Sampling the PBL more densely over those regions is likely to increase transport-driven uncertainty in fluxes. This is consistent with the hypothesis of Rayner and O'Brien (2001). However, over

some other regions, such as boreal Eurasia and tropical South America, the IS-LNLG spread is much smaller than the IS spread, suggesting that over those regions, the reduction in uncertainty going from IS to LNLG is primarily due to the more uniform spatial coverage and not due to total column sampling. In fact, over tropical South America the IS-LNLG spread is smaller than the LNLG spread, suggesting that the transport error in the total column is larger than that in the PBL. Finally, over regions such as Europe, the ordering of IS, IS-LNLG and LNLG uncertainties suggests that the reduction in uncertainty

in going from IS to LNLG is partly due to the more spatially uniform coverage and partly due to total column sampling.

Over some ocean regions such as the temperate North Pacific and South Atlantic, the IS-OG spread is larger than the OG spread, suggesting that modeling the PBL is more uncertain than modeling the total column over those regions. However, the opposite is true over several other ocean regions, such as the temperate North Atlantic and South Pacific. Thus, the hypothesis of Rayner and O'Brien (2001) cannot be said to hold over most ocean regions. Finally, one striking features of ocean fluxes in

figure 4 is worth pointing out here. The transport-derived uncertainty for IS-LNLG estimates is often the largest among all data streams, which leads to a large uncertainty in the global land/ocean partitioning using the IS-LNLG network. This suggests that increasing the PBL sampling only over land – where the transport models disagree more – is likely to worsen ocean flux estimates in the presence of imperfect transport models.

The global uptake, and its partitioning between land and ocean, or Northern and Southern Hemispheres, are less uncertain

for $XCO_2$ assimilations than for in situ $CO_2$ assimilations IS and MBL. Looking at the IS-LNLG and IS-OG inversions, we conclude that the improvement in the global budget and its north-south partitioning is likely due to a more uniform spatial coverage, while the improvement in land-ocean partitioning is likely due to the total column nature of the OCO-2 pseudo-data. Partitioning the budget in zonal bands, i.e, northern extra-tropics, tropics and southern extra-tropics, has (roughly) the same uncertainty across all inversions. This is likely due to the fast zonal flow in the free troposphere, which ensures that surface

flux signals missed by one set of measurements – perhaps due to imperfect transport – are seen by other measurements in the same zonal band.

Traditionally, inversions of surface $CO_2$ data have had larger uncertainty in tropical flux estimates compared to Northern Temperate regions, stemming from the sparse observational coverage in the tropics (Peylin et al., 2013). The larger interannual variability of the tropical flux, seen by several inversion studies including Baker et al. (2006) and Peylin et al. (2013), is also

ascribed partly to the higher uncertainty in tropical flux estimates. In contrast, the uncertainty in flux estimates stemming from

uncertainties in modeled transport do not have the same correlation with observational coverage. For inversions with in situ data, the relatively well-covered regions of North American temperate and Europe show the same transport-derived uncertainty as the poorly covered regions of Temperate South America and Tropical Asia (figure 4). In general, we do not find that the uncertainties in flux estimates due to transport model errors are lower over the northern temperate latitudes than over less

measured tropical and southern temperate areas.

One final noteworthy aspect of the flux estimates of figure 4 is that for some regions (such as temperate South America, Atlantic Tropics, Southern Ocean, South Indian temperate, tropical oceans, Indian ocean, the southern extra-tropics and southern extra-tropical land), the range of in situ flux estimates does not overlap with the range of LN, LG, or LNLG (and sometimes OG) flux estimates. For some other regions such as the Indian Ocean and the Southern Ocean, there is no overlap between the

OCO-2 land (LN, LG, LNLG) and ocean (OG) estimates. Since there are no biases between the IS, OCO-2 land and ocean pseudo-data, these flux differences suggest that spatiotemporal coverage differences between different observation networks and OCO-2 sampling modes can lead to flux differences that are larger than uncertainties due to transport.

## 4.4 Monthly fluxes

Figures 5 and 6 show the monthly flux estimates for 2015 from TRANSCOM-like land and ocean regions. As before, only

the spread across the pseudo-data generated by the five transport models is shown for visual clarity. The reduced sensitivity of OCO-2 pseudo-data inversions to transport model uncertainty is obvious for most months over both land and ocean regions. As before, this reduced sensitivity is from a combination of two factors, (a) spatially uniform coverage of OCO-2 compared to the in situ network, and (b) the assimilation of column average $XCO_2$ as opposed to PBL $CO_2$. The relative importance of the two factors – as gauged by the relative sizes of the bars between the OCO-2 (LN, LG, LNLG, OG), real in situ (IS) and

hypothetical in situ (IS-LNLG, IS-OG) data streams in figures 5 and 6 – varies by region and season. For example, in October in sub-Saharan Africa, going from the sparse IS network to the more uniform IS-LNLG network reduces the flux uncertainty significantly, but going from PBL measurements (IS-LNLG) to the total column (LNLG) does not reduce the uncertainty further. To contrast, over the same region in December, the increased PBL sampling of the IS-LNLG network inflates the flux uncertainty compared to the IS network, while going from PBL sampling (IS-LNLG) to the total column (LNLG) brings that

uncertainty down significantly. In general, over most land regions and most months, given OCO-2's spatiotemporal sampling, assimilating total column $CO_2$ (LNLG) results in equal or lower transport-driven uncertainty than assimilating PBL $CO_2$ (IS-LNLG). The same relationship holds between IS-OG and OG inversions over ocean regions with a few exceptions (e.g., South Indian temperate in June and July). However, the relationship between the real (IS) and hypothetical (IS-LNLG, IS-OG) networks is less general, and reflects the impact of different sampling.

The transport-derived uncertainty in monthly fluxes has clear seasonality over most land and ocean regions. In general, over temperate and boreal land regions, the uncertainty is higher in the summer than in the winter, likely due to stronger convective transport and higher horizontal wind shear in the summer months. Temperate oceans sometimes display the opposite behavior (e.g., temperate North Atlantic and North Pacific), whereby transport-driven uncertainty is lower in the summer and higher in the winter. This is likely because advective and not convective transport uncertainty is the dominant uncertainty over oceans.

Over the tropics the distinction is less clear cut, with no clear commonality between Tropical Asia and Tropical South America. Over the Tropical Indian ocean, the uncertainty is lowest in the last third of the year, whereas in the Tropical Pacific, the uncertainty is lowest in the middle of the year.

Over certain ocean regions (e.g., Atlantic Tropics, East Pacific Tropics, South Indian Temperate, Southern Ocean), the range of monthly fluxes obtained from synthetic $XCO_2$ over land (LN, LG and LNLG) often does not overlap at all with the range obtained from either the ocean data (OG) or in situ data (IS). Sometimes, the OCO-2 land pseudo-data inversions overlap with the ocean pseudo-data inversions but not with the true fluxes (e.g., temperate North Atlantic and North Pacific). Since there are no coherent biases in OCO-2 pseudo-data in these synthetic data experiments, the differences between land and ocean $XCO_2$ inversions, or between either set and the true fluxes, can only be due to differences in sampling the same $CO_2$ field with different sets of sampling times and locations. These sampling differences can lead to flux differences that are larger than the transport-driven uncertainty in fluxes. As noted earlier, this implies that in real data inversions biases can appear between land and ocean $XCO_2$ inversions, or between OCO-2 and in situ inversions, purely due to an imperfect transport model sampling the same field according to different sampling patterns. This can, for example, lead to biased flux estimates when ocean fluxes are inferred using OCO-2 land soundings, even when the retrievals are unbiased.

## 5  Conclusions

In this work, we have used five different transport models in an OSSE to estimate the uncertainty in inversion-derived flux estimates due to the uncertainty of the modeled transport in flux inversions. The five transport models were driven by four different state-of-the-art reanalyzed meteorological datasets that are commonly used in the flux inversion community, and therefore could be expected to span the spectrum of transport model behavior. In the OSSE, we created synthetic in situ and column $CO_2$ measurements by running the five transport models forward with the same boundary conditions and then assimilated those measurements in a single flux inversion system. The spread in the flux estimates was therefore purely due to the spread among the five transport models. We tested this setup for different sampling protocols: (a) an in situ set corresponding to NOAA's present-day cooperative air sampling network, (b) an in situ set of mostly background sites corresponding to the network used by Baker et al. (2006) for the TRANSCOM 3 model intercomparison experiment, (c) a set of $XCO_2$ measurements corresponding to OCO-2 land nadir, land glint and ocean glint soundings, convolved with corresponding OCO-2 averaging kernels and priors, and (d) a set of in situ samples within the PBL at the times and locations of OCO-2 land and ocean soundings. This allowed us to test the interaction of imperfect transport, observational coverage, and the assimilation of column versus PBL mole fractions. Our use of the OCO-2 data – both the temporal averaging and the errors on those averages – followed the current protocol used by OCO-2 flux modelers, and therefore our results should be directly usable by the modelers to draw conclusions about their real data inversions. There are four important take home messages from this work that we would like to convey.

## 5.1 MBL vs IS

A comparison of the spread of flux estimates from the MBL and IS inversions suggests that the added coverage from mostly continental sites on top of the mostly background network considered by Baker et al. (2006) can reduce transport-induced uncertainty over land regions, despite the uncertainty in transport over continents. This is likely due to the added observations averaging out some of the transport variability. The added coverage has minimal or negative benefit in reducing transport-induced uncertainty of ocean flux estimates, and estimates over zonal bands, except for the Pacific ocean and its temperate and tropical subdivisions.

## 5.2 Geographical distribution of transport uncertainty

For inversions of in situ data, flux estimates over the tropics have been historically less certain than estimates over the northern temperate regions, owing to lower observational coverage over the former. In previous work, the uncertainty of fluxes purely due to transport was also found to be slightly higher over tropical regions than over extra-tropical regions (Baker et al., 2006). However, in this work, we see that that demarcation does not hold for flux uncertainty stemming from transport model uncertainty. For example, the spread among IS inversions over Temperate North America or Europe in figure 4 is as large as their spread over Tropical Asia or Temperate South America respectively, despite the first two being much better covered with $CO_2$ samples.

## 5.3 Column vs PBL $CO_2$

Rayner and O'Brien (2001) had hypothesized that inversions of column average $CO_2$ may be less sensitive to vertical transport errors than PBL $CO_2$, since redistribution of $CO_2$ in the vertical does not change the column average. However, the variation of column $CO_2$ due to fluxes is also much smaller than in the PBL. The transport model sensitivity of column $CO_2$ inversions depends on the balance between this smaller flux signal and smaller transport error. In our experiments, we see that over TRANSCOM-scale and larger land regions (except tropical South America), inversions using column $CO_2$ data over land (LNLG) are indeed less sensitive to transport errors than inversions using PBL $CO_2$ at the same locations and times (IS-LNLG). Over TRANSCOM-scale ocean regions, however, the picture is more ambiguous, as several regions (e.g., Atlantic Ocean, South Pacific temperate, North Atlantic temperate, Southern Ocean) display a smaller uncertainty when assimilating PBL $CO_2$ (IS-OG) than column $CO_2$ (OG). This is likely because the uncertainty in convective transport over oceans is smaller than on land. The global budget and the partitioning across zonal bands are constrained equally well by column and PBL $CO_2$ samples, provided they have the same spatiotemporal coverage. The partitioning across land/ocean boundaries is noticeably more uncertain when using PBL samples over land than column samples, likely because vertical transport differences near the surface are larger over land than oceans.

It should be noted here that the low sensitivity of column measurements to PBL $CO_2$ variations is often considered a weakness, since surface flux signals are the largest in the PBL. Efforts are currently underway to construct active remote sensing instruments that are preferentially sensitive to the lower troposphere (Wang et al., 2014). Our OSSEs suggest that

were such an instrument to be deployed, the uncertainty of surface flux estimates derived from that instrument might very well be larger than from an OCO-2-like column $CO_2$ instrument due to transport model uncertainty near the surface. In the long term, significant improvement in transport modeling will be needed to benefit from a remote sensing instrument preferentially sensitive to near-surface $CO_2$.

## 5.4 Impact of coverage

In our synthetic data inversions, the difference between the fluxes inferred from the same forward model run but different sampling strategies is purely due to the interaction between non-ideal transport and data coverage, and not because of biases between the different samples. Despite this lack of bias, there are several regions where the entire spread of flux estimates across the five forward models has no overlap between certain types of data, or with the truth. For example, LN, LG and LNLG annual flux estimates from the Indian ocean have no overlap with either IS or OG estimate or the truth, while $XCO_2$ estimates of temperate South American fluxes are completely detached from all IS estimates. This effect is even more pronounced for monthly flux estimates. This suggests that in the presence of imperfect transport and no measurement bias, different coverage and sampling can generate biases in flux estimates that are larger than their uncertainty due to transport. We should therefore avoid infering, say, oceanic fluxes by using only OCO-2 land soundings.

## 6 Applicability of our work and future steps

While we have not used any real in situ or OCO-2 data in this work, the transport-driven uncertainty estimates we have presented can be used by other inverse modeling studies to test the robustness of their conclusions, when using a similar network of in situ and column $CO_2$ measurements. In future inversion intercomparisons along the lines of Houweling et al. (2015) and Peylin et al. (2013), which aggregate multiple model results, our uncertainty estimates can be used to infer whether the inter-model spread is driven primarily by transport model spread or by non-transport factors such as data selection and inversion methodology. We also plan to extend our work to multiple years to answer the question of whether the interannual variability (IAV) of flux estimates are more robust to differences in modeled transport than individual years. Baker et al. (2006) considered the same question for in situ data, but did not have IAV in their meteorology. By extending our study to multiple years in the future, we will be able to separate out the impact of transport model differences on the IAV for different sampling networks and observing platforms.

*Code and data availability.* All inversions for this work were performed in TM5 4DVAR, available publicly at https://sourceforge.net/projects/tm5. The OCO-2 soundings and their quality flags used to sample the models were obtained from https://disc.gsfc.nasa.gov/uui/datasets/OCO-2_L2_Standard_V7r/summary. The in situ sampling locations and times for sampling the models were obtained from NOAA's ObsPack portal at https://www.esrl.noaa.gov/gmd/ccgg/obspack. The times and locations of JR-STATION $CO_2$ samples were obtained from the National Institute for Environmental Studies at http://www.cger.nies.go.jp/en/climate/pj1/tower/.

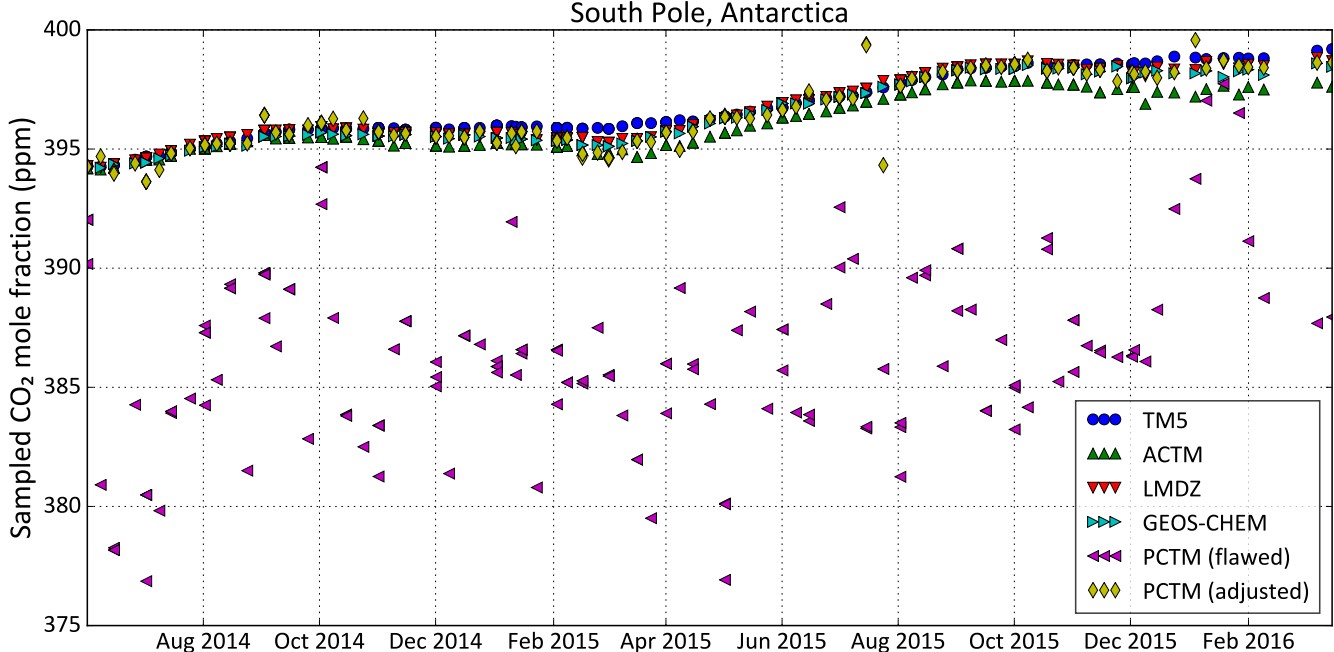

**Figure A1.** The modeled time series of the $CO_2$ mole fraction at NOAA flask sampling times at the South Pole station from all the models. PCTM ("PCTM (flawed)" here) is seen to have a problem, giving unrealistically low $CO_2$ values with unrealistically high variability. Moving the sampling site north by $2°$ along the Greenwich meridian, just for PCTM, greatly alleviates the problem ("PCTM (adjusted)").

## Appendix A: Adjusting PCTM mole fractions at South Pole

During this analysis, we discovered that the PCTM $CO_2$ field produced by our version of PCTM had a problem at the South Pole (SPO). There were low values of modeled $CO_2$ mole fraction high over SPO, which were propagating down over the sampling site and out over the Ross ice shelf. This caused unrealistically low modeled values and unrealistically high variations of $CO_2$ in PCTM at the SPO sampling site. Lacking a fix for this transport model artifact, we moved the SPO sampling site $2°$ north along the Greenwich meridian, which greatly reduced the problem. The time series of modeled $CO_2$ from all the models at the NOAA flask sampling times, along with the fixed sampling in PCTM, is shown in figure A1. We used this modified sampling of PCTM at SPO in this work. Until this bug is fixed, real data inversions with PCTM will use this or a similar modified sampling scheme at SPO as well.

## Appendix B: Maps of transport differences

Figure 3 showed the temporal evolution of the zonal average difference between each transport model and the model median. In figures B1 and B2, we show how that difference is distributed geographically in summer, winter and the annual average. The method of constructing these is exactly the same as for figure 3. All modeled $CO_2$ fields were mapped to a global $1°×1°$ grid

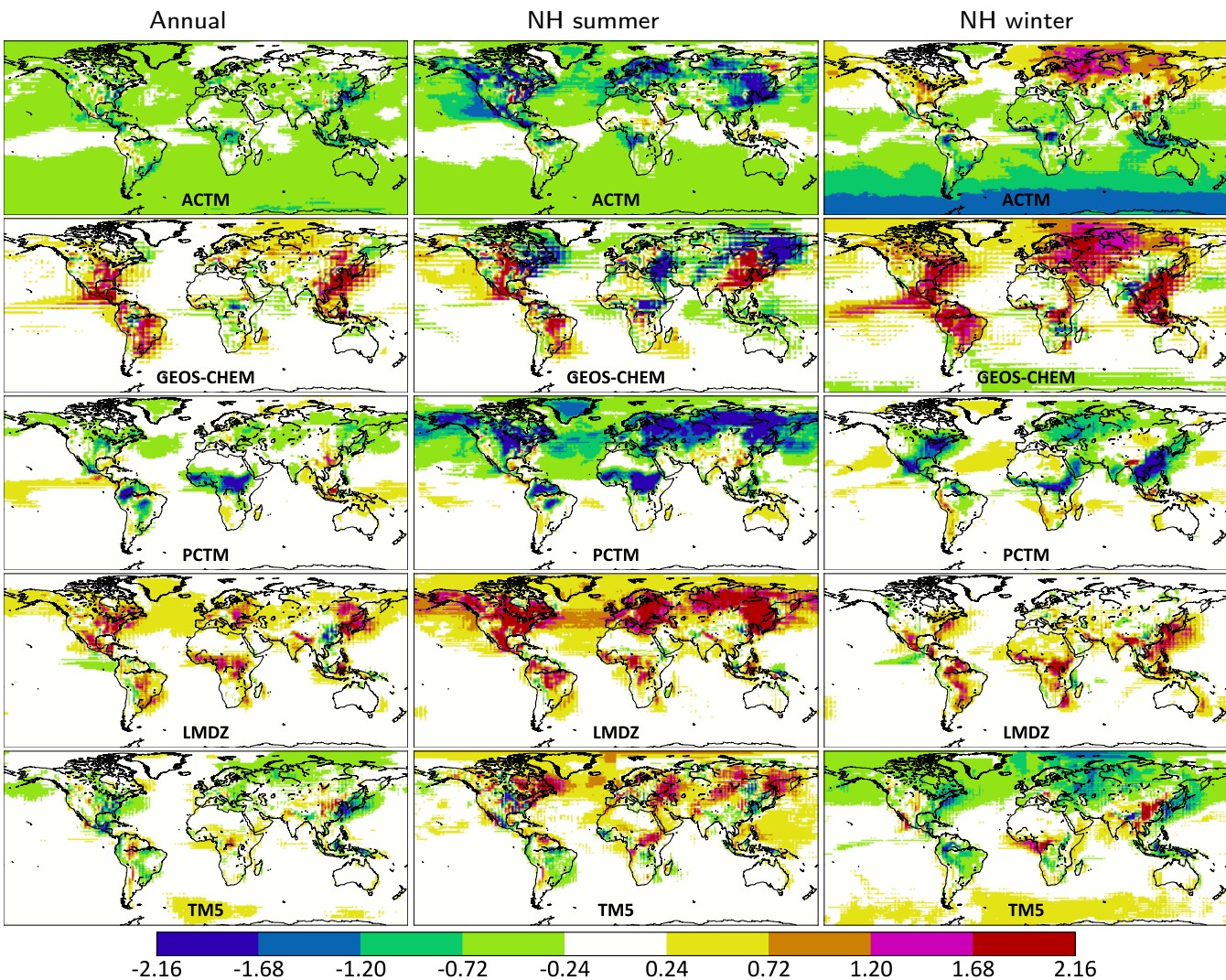

**Figure B1.** The difference between each model (ACTM, LMDZ, GEOS Chem, PCTM and TM5) and the cross-model median at 1:30 PM local time in the lowest 150 hPa, which is an approximation for the planetary boundary layer (PBL). The left column shows the difference averaged over all of 2015, the middle column is averaged over northern hemisphere summer months (Jun–Aug 2015), and the right column is averaged over northern hemisphere winter months (Dec 2015 to Feb 2016). Differences are shown in ppm $CO_2$.

while conserving mass. Since the models had varying resolutions and grid registrations, this resulted in unavoidable checkered patterns in the differences in figures B1 and B2. That, however, did not impact the large scale model to model differences shown. The color scale of figure B2 covers half the range of figure B1, since variations in the PBL are much larger than variations in the column.

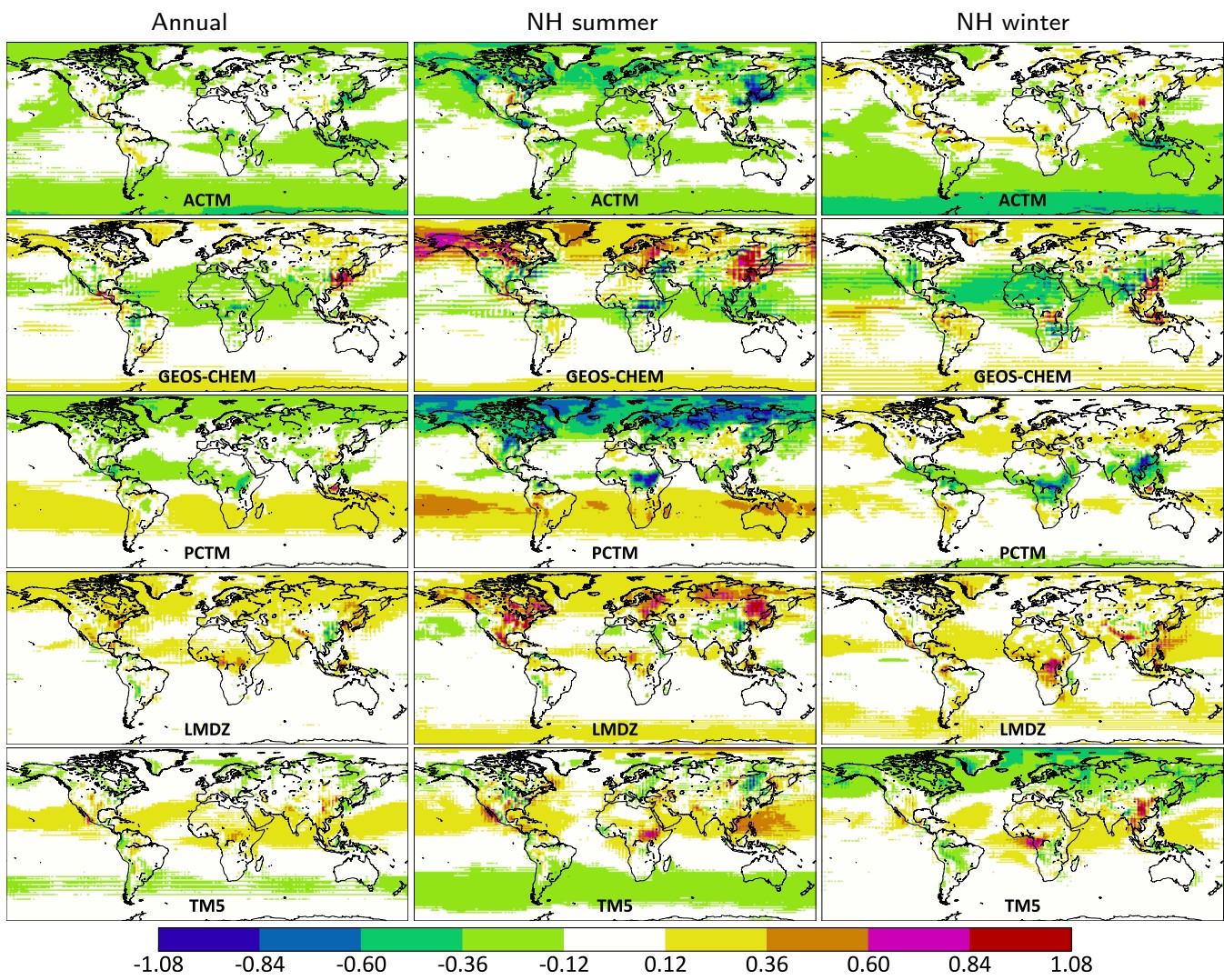

**Figure B2.** Same as figure B1, except averaged over the total column. The dynamic range here is half that of figure B1, since transport differences in the total column signal are smaller than in the PBL signal.

## Appendix C:  Spread in annual flux estimates as a function of the pseudo-data stream

Figure 4 displays the spread in annual flux estimates over various geographical regions from assimilating pseudo-data generated by the five forward models. Table C1 tabulates these spreads (minimum to maximum) for each region and choice of assimilated pseudo-data stream.

## Appendix D:  Annual flux estimates differentiated by forward model

In figure 4, the range of flux estimates for each data stream is shown, without distinguishing the flux estimates stemming from different forward models. Here, for the sake of completeness, we give the estimates from pseudo-data generated by each of the five models. In the plots below, different colored bars correspond to different synthetic data streams, while different marker shapes (such as square for TM5 and upright triangle for ACTM) correspond to the different transport models used to generate the synthetic data.

In figure D1, the TM5 symbols represent a "perfect transport" case, meaning the synthetic observations were generated and assimilated with the same transport model. Therefore, the difference between TM5 and Truth in the figure represents the balance between $S_a$ and $S_\epsilon$ in TM5 4DVAR, and a smaller difference from a different data stream (such as LMDZ with IS data over tropical land) is purely due to chance. It should also be noted that our goal in presenting the different models together in figure D1 is not to evaluate model performance by their proximity to either the Truth or perfect transport (TM5) results, but to evaluate the spread across different models used to generate the synthetic data, and how that spread varies with sampling and coverage.

*Author contributions.*  S. Basu wrote the paper with contributions from the other authors. ACTM, LMDZ, GEOS-Chem, PCTM and TM5 forward runs were performed by, respectively, P. Patra, F. Chevallier, J. Liu, D. Baker and S. Basu. Times, locations and uncertainties of 10 s average OCO-2 soundings were prepared by D. Baker. All inversions were done by S. Basu using TM5 4DVAR. J. Miller provided overall scientific oversight and guidance.

*Competing interests.*  We declare no competing interests.

*Acknowledgements.*  We would like to thank Kenneth Schuldt for preparing in situ $CO_2$ observations in NOAA's ObsPack format, which was used to derive the times and locations of in situ samples for our synthetic data sets. S. Basu and J. Miller would like to acknowledge support from NASA grant NNX15AH01G for the OCO-2 Science Team. F. Chevallier was funded by the Copernicus Atmosphere Monitoring Service, implemented by the European Centre for Medium-Range Weather Forecasts (ECMWF) on behalf of the European Commission. A portion of this research was carried out at the Jet Propulsion Laboratory, California Institute of Technology, under a contract with NASA.

**Table C1.** The spread in annual flux estimates for 2015 across the five forward models. These are the vertical extents of the colored bars in figure 4. The true and prior fluxes (white circle and grey diamond in figure 4) are also included as the last two columns of the table. All numbers are in PgC/year.

| Region | MBL | IS | LN | LG | LNLG | OG | IS-LNLG | IS-OG | True | Prior |
|---|---|---|---|---|---|---|---|---|---|---|
| North America | 1.86 | 0.83 | 0.34 | 0.24 | 0.23 | 0.42 | 1.31 | 1.23 | 0.22 | −0.55 |
| South America | 2.76 | 2.01 | 0.65 | 0.50 | 0.48 | 0.47 | 1.24 | 1.57 | 0.69 | 0.15 |
| Africa | 0.93 | 1.11 | 0.45 | 0.42 | 0.41 | 0.57 | 1.47 | 1.28 | −1.48 | −0.35 |
| Asia | 1.69 | 1.28 | 0.96 | 0.97 | 1.09 | 0.75 | 2.71 | 1.65 | −1.06 | −1.40 |
| North American boreal | 0.90 | 0.41 | 0.24 | 0.36 | 0.31 | 0.45 | 0.24 | 0.47 | −1.05 | −0.19 |
| North American temperate | 1.67 | 1.03 | 0.16 | 0.48 | 0.18 | 0.40 | 1.26 | 0.82 | 1.15 | −0.37 |
| South American tropical | 2.32 | 1.61 | 0.72 | 0.55 | 0.61 | 0.51 | 0.36 | 1.10 | −0.25 | 0.35 |
| South American temperate | 1.02 | 0.94 | 0.44 | 0.35 | 0.42 | 0.48 | 1.18 | 0.69 | 1.07 | −0.17 |
| Saharan Africa | 0.55 | 0.60 | 0.09 | 0.09 | 0.10 | 0.13 | 0.43 | 0.33 | −0.21 | −0.07 |
| Sub-Saharan Africa | 0.90 | 0.95 | 0.53 | 0.48 | 0.51 | 0.54 | 1.13 | 0.99 | −1.28 | −0.28 |
| Eurasian boreal | 1.95 | 1.28 | 0.29 | 0.27 | 0.33 | 0.35 | 0.61 | 1.39 | −0.86 | −1.06 |
| Eurasian temperate | 1.12 | 1.20 | 0.80 | 0.60 | 0.84 | 0.67 | 1.51 | 1.80 | −0.32 | −0.57 |
| Tropical Asia | 1.27 | 1.07 | 0.32 | 0.37 | 0.37 | 0.45 | 1.11 | 1.69 | 0.09 | 0.20 |
| Europe | 1.52 | 1.09 | 0.25 | 0.29 | 0.30 | 0.36 | 0.72 | 0.64 | −0.56 | 0.03 |
| Australia | 0.46 | 0.37 | 0.17 | 0.18 | 0.19 | 0.20 | 0.29 | 0.14 | −0.22 | 0.07 |
| Pacific Ocean | 1.98 | 1.16 | 0.20 | 0.27 | 0.28 | 0.84 | 2.76 | 1.12 | 0.07 | −0.60 |
| Atlantic Ocean | 0.55 | 0.67 | 0.80 | 0.65 | 1.14 | 0.86 | 1.75 | 0.33 | −0.38 | −0.58 |
| Indian Ocean | 0.28 | 0.21 | 0.20 | 0.39 | 0.28 | 0.23 | 0.60 | 0.21 | −0.22 | −0.52 |
| North Pacific temperate | 0.89 | 0.68 | 0.47 | 0.45 | 0.50 | 0.41 | 2.23 | 1.03 | −0.39 | −0.55 |
| West Pacific tropics | 0.17 | 0.13 | 0.02 | 0.02 | 0.03 | 0.05 | 0.19 | 0.11 | 0.04 | 0.04 |
| East Pacific tropics | 0.58 | 0.48 | 0.34 | 0.32 | 0.50 | 0.22 | 0.88 | 0.25 | 0.78 | 0.41 |
| South Pacific temperate | 0.62 | 0.51 | 0.28 | 0.29 | 0.41 | 0.35 | 0.78 | 0.20 | −0.36 | −0.49 |
| Northern Ocean | 0.17 | 0.26 | 0.10 | 0.05 | 0.07 | 0.11 | 0.07 | 0.25 | −0.17 | −0.19 |
| North Atlantic temperate | 0.41 | 0.84 | 0.67 | 0.56 | 0.89 | 0.81 | 1.45 | 0.23 | −0.32 | −0.27 |
| Atlantic tropics | 0.18 | 0.10 | 0.12 | 0.11 | 0.17 | 0.23 | 0.20 | 0.20 | 0.36 | 0.12 |
| South Atlantic temperate | 0.44 | 0.42 | 0.13 | 0.15 | 0.22 | 0.31 | 0.49 | 0.50 | −0.41 | −0.43 |
| Southern Ocean | 0.32 | 0.34 | 0.14 | 0.19 | 0.17 | 0.58 | 0.47 | 0.40 | −0.41 | −0.37 |
| Indian tropical | 0.13 | 0.11 | 0.05 | 0.06 | 0.08 | 0.11 | 0.19 | 0.22 | 0.10 | 0.14 |
| South Indian temperate | 0.26 | 0.28 | 0.18 | 0.34 | 0.24 | 0.26 | 0.44 | 0.30 | −0.31 | −0.66 |
| Northern extra-Tropical land | 1.08 | 1.39 | 1.11 | 1.00 | 1.38 | 1.27 | 3.38 | 1.43 | −2.20 | −2.13 |
| Northern extra-Tropical ocean | 0.71 | 1.07 | 0.95 | 0.79 | 1.05 | 0.51 | 3.12 | 0.94 | −0.93 | −0.93 |
| Northern extra-Tropics | 1.03 | 0.95 | 1.01 | 0.91 | 1.13 | 1.24 | 0.94 | 0.90 | −3.17 | −3.07 |
| Tropical land | 2.12 | 2.10 | 0.97 | 1.08 | 0.93 | 1.06 | 1.74 | 1.16 | −0.83 | 0.22 |
| Tropical ocean | 1.02 | 0.82 | 0.48 | 0.51 | 0.69 | 0.39 | 1.63 | 0.70 | 1.33 | 0.44 |
| Tropics | 1.49 | 1.54 | 1.43 | 1.44 | 1.50 | 1.39 | 1.48 | 0.99 | 0.45 | 0.67 |
| Southern extra-Tropical land | 0.90 | 0.53 | 0.40 | 0.37 | 0.41 | 0.54 | 0.60 | 0.32 | 0.69 | −0.15 |
| Southern extra-Tropical ocean | 0.54 | 0.53 | 0.36 | 0.53 | 0.43 | 0.78 | 0.44 | 0.97 | −1.51 | −1.78 |
| Southern extra-Tropics | 0.49 | 0.58 | 0.72 | 0.81 | 0.73 | 0.95 | 0.95 | 0.98 | −0.92 | −1.92 |
| Global land | 2.23 | 1.83 | 0.91 | 0.84 | 1.17 | 0.52 | 4.84 | 0.93 | −2.52 | −2.06 |
| Global ocean | 1.57 | 0.94 | 1.08 | 1.02 | 1.22 | 0.23 | 4.87 | 0.99 | −1.12 | −2.26 |
| Northern Hemisphere | 1.99 | 1.78 | 0.64 | 0.73 | 0.66 | 0.83 | 0.69 | 1.13 | −3.71 | −3.08 |
| Southern Hemisphere | 0.56 | 1.41 | 0.81 | 0.85 | 0.82 | 0.98 | 0.97 | 1.07 | 0.06 | −1.24 |
| Globe | 1.71 | 1.51 | 0.22 | 0.24 | 0.24 | 0.29 | 0.33 | 0.29 | −3.64 | −4.32 |

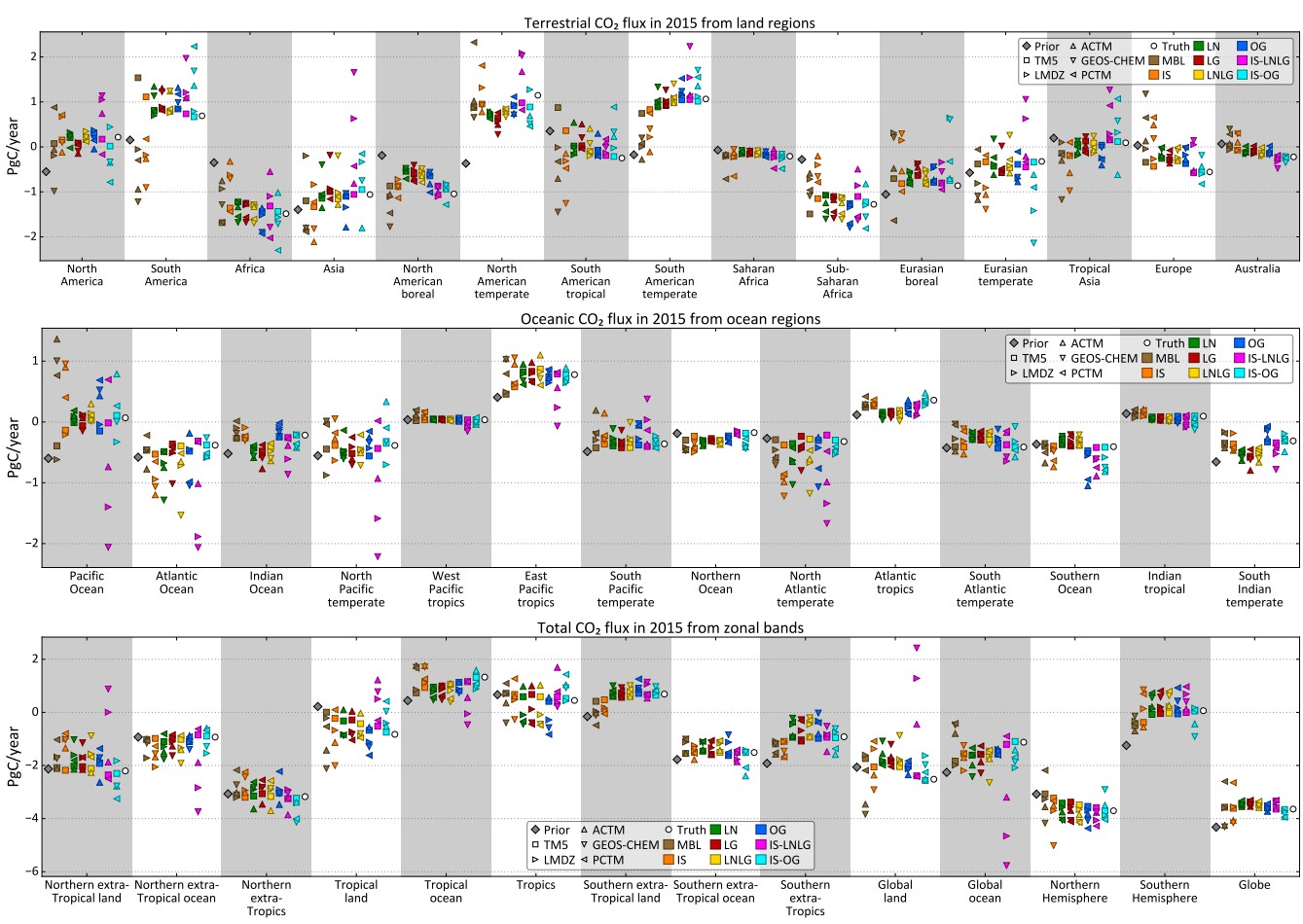

**Figure D1.** Annual flux estimates from TRANSCOM-like regions (top), zonal bands (middle) and large land and ocean regions (bottom). The different colors correspond to different synthetic data streams assimilated, IS = in situ, LN = land nadir, LG = land glint, OG = ocean glint, LNLG = LN + LG. The IS-LNLG and IS-OG are hypothetical PBL networks described in § 2.3.3. For each color, the different symbols denote the forward model used to produce the pseudo-data that was assimilated by TM5 4DVAR.

J. Liu was supported by NASA grant 13-CMS13-0025. D. Baker was supported by NASA grant NNX14AO77. P. Patra was supported by the Environment Research and Technology Development Fund (2-1701) of the Ministry of the Environment, Japan. GEOS-Chem forward runs were performed on the Pleiades cluster at the NASA Advanced Supercomputing (NAS) center. ACTM forward runs were performed at JAMSTEC's supercomputing facility in Yokohama. All inversions for this work were performed on the Discover cluster at the NASA Center for Climate Simulations (NCCS). The OCO-2 sounding locations, times, and averaging kernels were obtained from data that is publicly available from the Goddard Earth Science Data and Information Services Center (GES-DISC), https://disc.gsfc.nasa.gov/datacollection/ OCO2_L2_Standard_7.html.

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
