# Peer review of "The Impact of Transport Model Differences on CO2 Surface Flux Estimates from OCO-2 Retrievals of Column Average CO2"

_Atmospheric Chemistry and Physics, 2017_

## Referee Comment (RC1) · Anonymous Referee #1 · 7 Feb 2018

This study reports observing system simulation experiments aiming at assessing the impact of atmospheric transport model uncertainties on inverse modelling estimated sources and sinks of carbon dioxide using surface and satellite measurements. The results confirm the outcome of similar studies conducted in the past indicating that transport models are a potentially performance limiting component in the translation of atmospheric measurements to surface fluxes. This study takes a further step to compare the role of transport in inversions using either surface or satellite data. Conclusions are drawn on the importance of homogeneous sampling and the sensitivity of the use of satellite or surface measurements to important uncertain factors in the transport models. In my opinion, as explained below, these conclusions need to be

[Figure]

better supported by the results (or the way in which the results are assessed) to make this study acceptable for publication.

GENERAL COMMENTS

Sections 4.2 and 4.4, which present the experimental evidence for the conclusions drawn in section 5, provide almost no numbers. The discussion is on a very qualitative level. This makes it difficult to judge the significance of the conclusions and will make it difficult for future studies to compare results to what was found here. Further effort is needed to repair this and to ensure that the main conclusions are supported by actual numbers and corresponding uncertainties. To assess significance requires being able to compare the impacts of transport model uncertainty on the fluxes to the uncertainty of the estimated fluxes themselves. Right now, it is difficult to judge whether the impacts are in the noise of the estimates (probably not), or limiting the overall performance.

In the discussion about the main factors explaining the impact of transport model uncertainties on the estimated fluxes using surface or satellite data, suggestions are made about the role of horizontal and vertical sampling. Since several factors vary at the same time, I don't see how the importance of these impacts can be addressed in isolation. It would require a total column measurement at the times and locations of the surface measurements or satellite measurements being sensitive only to the surface. Even then it not easy to account for differences in measurement constraints on the inversion influencing the impact of transport model uncertainties on the optimized fluxes. In the discussion, it should be made clearer that the experiments do not provide direct evidence for specific causes.

SPECIFIC COMMENTS

page 5, line 10: What motivates the chosen time span?

page 7, line 6: item 2: Is daytime sampling used for marine background sites also?

page 9, line 10: Why were posterior fluxes from CarbonTracker chosen as prior? They

are not independent from the data that are used to derived the truth ... to me it seems more logical to take the CarbonTracker prior. How consistent is the choice of prior covariances in this case?

page 10, line 12: What is Nret typically? Does epsilonˆ2/Nret yield a realistic systematic error?

page 11, line 5: Even if all the models had the same random noise added to the data, this would not have changed the inter model spread in the fluxes. However, if the importance of transport model uncertainty is assessed in relation to overall posterior flux uncertainty then measurement uncertainties do matter (whether or not you would add random noise to the data in this case depends on the method for calculating posterior flux uncertainties).

page 11, equation 6: So, in this case the difference between 2 models doesn't even depend on the choice of prior flux. This means that my earlier remark about the use of CarbonTracker posterior fluxes actually doesn't matter. It would still be useful to point out that the prior fluxes that are described in detail aren't really relevant to the problem .. well, they are to the extent that the a priori fluxes are used to define a priori flux uncertainties. Some further sentences clarifying this would be useful.

page 12, line 8: what do you mean by 'lateral' grid cell? Each individual grid cell?

page 14, line 17: 'due to chance' you mean 'due to differences in transport'?

page 19, line 25: but since the sampling is also very different between surface and satellite there is no way to isolate the impact of PBL versus total column.

page 22, line 21: I wonder if this difference between this study and Baker et al (2006) could be influenced by the choice of an El Nino year for the current inter-comparison, which may not be well representative of a typical year (so my earlier remark about justifying the chosen times window).

page 22, line 28: .. but could also be due to a more even sampling coverage.

TECHNICAL CORRECTIONS

page 4, line 26: 'initial' i.o. 'intial'

page 7, line 31: 'referred' i.o. 'refered'

caption figure 3: 'color bar' i.o. 'colorbar'

---

## Referee Comment (RC2) · Anonymous Referee #2 · 15 Mar 2018

The manuscript presents an assessment of errors in atmospheric transport modelling from the context of trace gas transport, with a particular focus on carbon dioxide. It is a synthetic data experiment, using realistic sampling, but no actual measurements. Several different transport models are used for forward simulations, to assess the divergence of simulated concentration values based on the same fluxes. The effect on inferred fluxes is also assessed, by using one of the transport models to invert the pseudomeasurements produced by the other transport models. In general, the approach is very similar to that of previous studies (Chevallier et al., 2010; Houweling et al., 2010; Locatelli et al., 2013), which the authors do cite. The main difference seems to be the focus on OCO-2 for the source of satellite sampling (in addition to in situ

measurements), and the separation of different measurement modes in the inversions (ocean glint, land glint, land nadir, etc.). This leads to the interesting finding that sampling differences can be larger than flux uncertainties due to transport differences. In general the paper is quite well written and presented, and appropriate for publication in ACP, once some concerns have been addressed.

Comments:

P12,L13-15: This claim is made repeatedly throughout the paper, that the larger differences in concentrations over land are because of larger differences in the vertical mixing over land. One would expect this to be true, one should also consider the fact that the spatiotemporal variability of the fluxes is also considerably higher over land, and what you're assessing is the differences in the concentration fields, not the air mass fluxes themselves. The oceans may show less agreement also because the flux pattern is comparatively heterogeneous there. (Consider the extreme case with a well-mixed atmosphere and zero fluxes: the differences in concentration space would be zero amongst the different transport models.) Please include this in your discussion, here and elsewhere.

P20,L12-14: Same thing. The flux variability is also higher in the summer. You're only looking at the tracer concentrations, which are a result of the mixing plus the fluxes themselves.

P12,from L16: In general the discussion around Figure 3 was very difficult to follow. What is meant by "venting"? This has to be defined. The sign of the fluxes changes over the year, so if a "higher venting" means more mixing with the free troposphere, this would change signs throughout the year (lower than the median in winter, higher than the median in summer). In this case, I think that TM5 shows higher venting than LMDZ over the northern hemisphere winter, doesn't it? And how much of this "venting" is simply a different PBL height? Or is something else meant here altogether? Later on it is mentioned that fluxes are vented to the south - so is this more a measure of interhemispheric and long-range transport? Or does the venting discussion include land-ocean longitudinal transport? If so, this can't be seen from the zonal plot presented here.

Please change either Figure B1 or Figure 3 to put the rows in the same order.

P12,L25: The reference to Boreal Eurasia: This can't be seen from the zonal plot, please refer to the figure in the appendix here.

P18,L15-16: You mention that the observation mode changed (from 16-day nadir-/glint-only cycles), but don't state exactly when, and if you can see any effect of this in the time series of your retrieved fluxes. Please add this information.

Minor technical comments:

P7,L28: "from assimilating a more limited, mostly background sites": Maybe missing "set of", or remove "a"?

Figure 4 (and other similar figures): It's a minor thing, but many in this community have the standard order of the TransCom regions (at least over land) more or less memorized. Why switch Europe and Australia? Sticking to the standard order would make it easier for the reader.

P20,L34: "the uncertainty in flux estimates due to transport model errors are lower ": subject and verb don't agree (are->is).

P21,L20-21: "the range of monthly fluxes obtained from synthetic XCO2 over land (LN, LG and LNLG) often do not overlap": subject and verb don't agree (do->does).

P22,L23-24: "the spread among IS inversions over Temperate North America and Europe in figure 4 are as large as their spreads over Tropical Asia and Temperate South America": again: (are->is), plus I would keep "spread" singular in the second instance as well.
* * *

---

## Author Comment (AC1) · 18 Apr 2018

We thank the referee for carefully reviewing our manuscript. We have accepted several of his suggestions, which we believe have significantly improved the manuscript. The referee had two major general comments, (i) the results presented in §4.2 and §4.4 were mostly qualitative, and (ii) it was difficult to draw conclusions about total column vs planetary boundary layer (PBL) measurements from the experiments because of differing spatiotemporal coverage.

Regarding the qualitative nature of results in §4, it is difficult to present general quantitative results in a work like this, because the impact of transport model uncertainty varies by region, time and data stream. Previous work that tackled the question of assessing transport model uncertainty, such as Locatelli et al (2013), also struggled with drawing quantitative yet general conclusions. The referee mentions comparing our transport-derived uncertainties with posterior uncertainties of the flux estimates. This is also not robust, since the posterior uncertainty from TM5 4DVAR is an overestimate (Meirink et al, 2008), a problem common to most iterative flux inversion techniques. Instead, this work considers the cross-model spread of OCO-2 inverse models (Crowell et al, 2018) a measure of the uncertainty of our knowledge of inversion-derived surface fluxes, and tries to estimate whether those uncertainties are consistent with what we would expect just from transport model uncertainty.

However, we agree that our existing results could be presented in a more quantitative form. We have added several tables in the text to facilitate this. Table 2 shows the number of observations assimilated from each data stream, Table 3 gives the uncertainty (spread across five transport models) in the global budget and its partitioning (land/ocean or latitude band), and Table C1 gives the uncertainty in the annual flux from all the geographical regions considered, along with the prior and true fluxes. Comparing the uncertainty with the true flux gives an idea of the significance of the uncertainty.

Regarding the differing coverage of OCO-2 and in situ measurements, this is a very good point. To make a clearer distinction between the impact of total column measurement (vs PBL) and the impact of a spatially distributed sampling pattern, we created two hypothetical in situ networks. IS-LNLG (IS-OG) had PBL samples 30m above ground level at the times and locations of all OCO-2 land (ocean) soundings used in the LNLG (OG) inversions. A comparison between LNLG and IS-LNLG (OG and IS-OG) inversions, therefore, reveal the impact of having total column vs PBL measurements over land (ocean), and not differences in spatiotemporal coverage. We have replaced our figures and tables to include these new (hypothetical) data streams, and have reworked our results and conclusions to incorporate the new results. The short summary is that over most land regions, total column samples do lower the transport-driven uncertainty in flux estimates compared to PBL samples. This holds less strictly over ocean regions, likely due to lower convective fluxes (and hence lower model to model differences). The land/ocean partitioning within a zonal band is more uncertain with land PBL samples, but the aggregate over the zonal band is not. Flying a remote sensing instrument with higher PBL sensitivity has been a goal of space-based greenhouse gas missions (Wang et al, 2014). Our results suggest that if such an instrument were to fly, the uncertainty in transport modeling would become a severe bottleneck, and considerable improvement in transport modeling would be needed before we could use such an instrument to improve on the precision of estimated surface fluxes.

Specific comments

"What motivates the chosen time span?"

The time span was motivated by two factors. (a) When this study was initiated (late 2016), OCO-2 data were available up to July 2016. Allowing for some spin-up and spin down, this left 2015 as the only full calendar year

we could address. (b) The initial goal of the OCO2 model intercomparison project (Crowell et al, 2018) was to perform inversions to estimate and compare flux estimates for 2015. Since this work was supposed to help them test the robustness of their conclusions, it made sense to perform our work over the same time period. Having said that, we certainly want to extend this work to at least three years in the near future, to study questions such as trend and interannual variability that cannot be addressed with one year's fluxes.

"Is daytime sampling used for marine background sites also?"

Yes, the majority of sites in the MBL network were sampled in the local mid-afternoon. Mountaintop sites such as Mauna Loa were sampled in the early morning in both the IS and MBL networks to reduce the possibility of updrafts. In fact, the samples in the MBL network are a subset of those in the IS network. We simply chose those samples in the IS network that belonged to sites used by Baker et al (2006).

"Why were posterior fluxes from CarbonTracker chosen as prior? They are not independent from the data that are used to derived the truth ... to me it seems more logical to take the CarbonTracker prior. How consistent is the choice of prior covariances in this case?"

The CT posterior was chosen as the prior for two reasons. (a) The CT prior does not have a net ecosystem sink, and using such as obviously biased prior guarantees a biased posterior in an inversion. This is why some long term inversions evaluate their fluxes with respect to fluxes that already guarantee the correct atmospheric $CO_2$ trend (Chevallier et al, 2010). (b) Several inversions in the OCO2 model intercomparison project (Crowell et al, 2018) is also using a climatological CT posterior as the prior flux, and our goal was to make our experiments maximally relevant to that effort. We also note here that the 2000-2015 average posterior, which we used as the prior, will have little information specifically from 2015 observations. In any case, our conclusions primarily concern transport-driven uncertainty, which is not expected to be sensitive to the choice of prior (a fact the referee notes later).

The question of the prior covariance is an interesting one. In our system we specify the prior error as a fraction of the CASA heterotrophic respiration and not the NEE. As such, it is not strongly coupled to our choice of the NEE; the fractional change in the NEE from prior to posterior is a very small change in comparison to the heterotrophic respiration. Moreover, since our prior uncertainty uses the same CASA vegetation map as CT, it is guaranteed to be large (small) where CT thinks there is a lot of (no) ecosystem activity.

"What is Nret typically? Does epsilon^2/Nret yield a realistic systematic error?"

$N_{ret}$ can be anywhere between 1 and 24. See Figure 1 for histograms of $N_{ret}$ over land and ocean. To answer the second part of the question, we plotted the standard deviation of $XCO_2$ coming from the retrievals in 1s bins in Figure 1. The purpose of the epsilon²/$N_{ret}$ is to prevent the value of this spread from getting too low, as might be the case

[Figure]

**Figure 1:** Frequency distribution of $N_{ret}$ over land (far left) and ocean (near left) over ten days chosen randomly from the OCO2 record. Over land (LN+LG), $N_{ret} = 1$ 19% of the time, $N_{ret} = 2$ 7.5% of the time, etc. The standard deviation of retrieved $XCO_2$ in 1s bins over land and ocean are shown in near right and far right respectively.

when there are only a couple of shots in the bin and they happen to have close to the same $XCO_2$ value (or the extreme case, when there is only a single shot, in which case the standard deviation is zero).

"Even if all the models had the same random noise added to the data, this would not have changed the inter model spread in the fluxes. However, if the importance of transport model uncertainty is assessed in relation to overall posterior flux uncertainty then measurement uncertainties do matter (whether or not you would add random noise to the data in this case depends on the method for calculating posterior flux uncertainties)."

This is true. However, as explained earlier, we are not comparing the transport model uncertainty to the analytical posterior flux uncertainty from any single inversion, because our system cannot provide a robust estimate of that. Rather, the goal is to use these transport uncertainties as guides when comparing flux estimates (not their uncertainties) from different assimilation systems. And usually an assimilation system does not add random noise to the measurements.

"So, in this case the difference between 2 models doesn't even depend on the choice of prior flux. This means that my earlier remark about the use of CarbonTracker posterior fluxes actually doesn't matter. It would still be useful to point out that the prior fluxes that are described in detail aren't really relevant to the problem.. well, they are to the extent that the a priori fluxes are used to define a priori flux uncertainties. Some further sentences clarifying this would be useful."

This is a good point, and we have added a sentence clarifying this.

"what do you mean by 'lateral' grid cell? Each individual grid cell?"

Yes. We had used the term 'lateral grid' to distinguish it from the 'vertical grid', but realize that that is clear enough from the context. We have deleted the word 'lateral' from that sentence.

"'due to chance' you mean 'due to differences in transport'?"

What we meant was that the proximity of the flux from the TM5 data stream to the true flux reflected the posterior uncertainty of our inversion system. If a different model's result happened to be closer to the true flux for some region, it should not be taken to mean that that model somehow provided "better" than perfect transport. However, we agree that meaning was not clear. We have changed that sentence to read "should not be interpreted as significant".

"but since the sampling is also very different between surface and satellite there is no way to isolate the impact of PBL versus total column."

See our description above of the hypothetical networks IS-LNLG and IS-OG. We have added several inversions with these networks in the revised version.

"I wonder if this difference between this study and Baker et al (2006) could be influenced by the choice of an El Nino year for the current inter-comparison, which may not be well representative of a typical year (so my earlier remark about justifying the chosen times window)."

That is possible, but it's not clear to us why tropical and temperate transport uncertainties would change differently in an El Nino year. This is definitely something to look at when we extend out study to multiple years.

"but could also be due to a more even sampling coverage."

This has now been addressed with our new model runs.

Technical corrections

All three technical corrections have been implemented.

References

D. F. Baker, R. M. Law, K. R. Gurney, P. Rayner, P. Peylin, A. S. Denning, P. Bousquet, L. Bruhwiler, Y.-H. Chen, P. Ciais, I. Y. Fung, M. Heimann, J. John, T. Maki, S. Maksyutov, K. Masarie, M. Prather, B. Pak, S. Taguchi, and Z. Zhu, "TransCom 3 inversion intercomparison: Impact of transport model errors on the interannual variability of regional CO2 fluxes, 1988-2003," Glob. Biogeochem. Cycles, vol. 20, no. 1, p. GB1002, Jan. 2006.

F. Chevallier, P. Ciais, T. J. Conway, T. Aalto, B. E. Anderson, P. Bousquet, E. G. Brunke, L. Ciattaglia, Y. Esaki, M. Fröhlich, A. Gomez, A. J. Gomez-Pelaez, L. Haszpra, P. B. Krummel, R. L. Langenfelds, M. Leuenberger, T. Machida, F. Maignan, H. Matsueda, J. A. Morguí, H. Mukai, T. Nakazawa, P. Peylin, M. Ramonet, L. Rivier, Y. Sawa, M. Schmidt, L. P. Steele, S. A. Vay, A. T. Vermeulen, S. Wofsy, and D. Worthy, "$CO_2$ surface fluxes at grid point scale estimated from a global 21 year reanalysis of atmospheric measurements," J. Geophys. Res., vol. 115, no. D21, pp. D21307–D21307, Nov. 2010.

S. Crowell, et al, "The 2015-2016 Carbon Cycle As Seen from OCO-2 and the Global In Situ Network", in prep (2018).

R. Locatelli, P. Bousquet, F. Chevallier, A. Fortems-Cheney, S. Szopa, M. Saunois, A. Agusti-Panareda, D. Bergmann, H. Bian, P. Cameron-Smith, M. P. Chipperfield, E. Gloor, S. Houweling, S. R. Kawa, M. Krol, P. K. Patra, R. G. Prinn, M. Rigby, R. Saito, and C. Wilson, "Impact of transport model errors on the global and regional methane emissions estimated by inverse modelling," Atmos. Chem. Phys., vol. 13, no. 19, pp. 9917–9937, Oct. 2013.

J. F. Meirink, P. Bergamaschi, and M. C. Krol, "Four-dimensional variational data assimilation for inverse modelling of atmospheric methane emissions: method and comparison with synthesis inversion," Atmos. Chem. Phys., vol. 8, pp. 6341–6353, Jun. 2008.

J. S. Wang, S. R. Kawa, J. Eluszkiewicz, D. F. Baker, M. Mountain, J. Henderson, T. Nehrkorn, and T. S. Zaccheo, "A regional $CO_2$ observing system simulation experiment for the ASCENDS satellite mission," Atmos. Chem. Phys., vol. 14, no. 23, pp. 12897–12914, 2014.

---

## Author Comment (AC2) · 18 Apr 2018

We thank the referee for carefully reading the manuscript and providing valuable suggestions. Please see our responses to the reviewer's comments below.

**P12, L13-15**

This is a fair point. While we do think that transport uncertainty (transport model spread) is higher over land, which leads to higher transport-driven uncertainty from land samples over land than from ocean samples over oceans, that point has not yet been demonstrated at this point in the manuscript. So we have added a sentence here to say that $CO_2$ differences are larger over land due to both transport and flux variability being higher over land.

**P20, L12-14**

We respectfully disagree with the referee here. In these lines we are talking about the uncertainty in inverted fluxes, not differences in the simulated $CO_2$ fields. We are referring to the higher spread in IS and MBL inversions over land as evidence that transport variability, at least on land, is higher in the PBL than in the total column. We would also like to refer the reviewer to the new IS-LNLG and IS-OG experiments that we have included in the revised manuscript, which explicitly tries to address the impact of coverage versus the impact of total column sampling.

**P12, from L16**

The referee is correct, we have not defined "venting", and it is an ambiguous term. Sometimes it means exchange between the PBL and the free troposphere, and sometimes it means inter-hemispheric exchange. We have removed all mention of "venting" and used more exact terms in the revised manuscript. Regarding the comparison between LMDZ and TM5 in NH winter, the referee is right. The surface signal is positive, so the model that has a faster (slower) PBL-FT exchange will have lower (higher) $CO_2$ near the surface. Since TM5 has lower $CO_2$ in the continental PBL in the NH winter, TM5's PBL-FT exchange must be higher than LMDZ's. We have corrected this in the revised manuscript.
In figure 3, the positions of PCTM and LMDZ have been exchanged to be consistent with figures B1 and B2, as per the referee's suggestion.

**P12, L25**

Reference to figure B1 added.

**P18, L15-16**

The 16-day nadir/glint mode lasted till early July 2015. This information has been added. However, there is no drastic change in the difference between fluxes from nadir and glint inversions after that date, so we only mention this as one of two possible reasons for why nadir and glint-derived fluxes may be different despite no relative coherent bias.

**P7, L28**

Added "set of".

**Figure 4 (and other similar figures)**

The referee is right that our order of regions is not the standard TRANSCOM region order. However, in Figure 4 we are not just presenting the TRANSCOM regions, so we would respectfully suggest that having

the same region order is not crucial. In figures 5 and 6 we **are** presenting only TRANSCOM regions (and their totals), so in those figures we have switched the order to conform to the standard TRANSCOM order.

Change "uncertainty" to "uncertainties" since we are referring to multiple regions.

Changed "do" to "does".

That sentence has been fixed, using the singular in all instances and a judicious application of "respectively".